 

# The COMA complex interacts with Cse4 and positions Sli15/Ipl1 at the budding yeast inner kinetochore

Josef Fischböck-Halwachs[1,2†], Sylvia Singh[1,2†], Mia Potocnjak[1,2†], Götz Hagemann[1,2], Victor Solis-Mezarino[1,2], Stephan Woike[1,2], Medini Ghodgaonkar-Steger[1,2], Florian Weissmann[3], Laura D Gallego[4], Julie Rojas[5], Jessica Andreani[6], Alwin Köhler[4], Franz Herzog[1,2]*

[1]Gene Center Munich, Department of Biochemistry, Ludwig-Maximilians-Universität München, Munich, Germany; [2]Department of Biochemistry, Ludwig-Maximilians-Universität München, Munich, Germany; [3]Research Institute of Molecular Pathology (IMP), Vienna Biocenter (VBC), Vienna, Austria; [4]Max F Perutz Laboratories, Medical University of Vienna, Vienna, Austria; [5]Laboratory of Chromosome Biology, Max Planck Institute of Biochemistry, Martinsried, Germany; [6]Institute for Integrative Biology of the Cell (I2BC), CEA, CNRS, Université Paris-Sud, Université Paris-Saclay, Gif-sur-Yvette, France

*For correspondence:
herzog@genzentrum.lmu.de

†These authors contributed equally to this work

**Competing interests:** The authors declare that no competing interests exist.

**Abstract** Kinetochores are macromolecular protein complexes at centromeres that ensure accurate chromosome segregation by attaching chromosomes to spindle microtubules and integrating safeguard mechanisms. The inner kinetochore is assembled on CENP-A nucleosomes and has been implicated in establishing a kinetochore-associated pool of Aurora B kinase, a chromosomal passenger complex (CPC) subunit, which is essential for chromosome biorientation. By performing crosslink-guided in vitro reconstitution of budding yeast kinetochore complexes we showed that the Ame1/Okp1[CENP-U/Q] heterodimer, which forms the COMA complex with Ctf19/Mcm21[CENP-P/O], selectively bound Cse4[CENP-A] nucleosomes through the Cse4 N-terminus. The Sli15/Ipl1[INCENP/Aurora-B] core-CPC interacted with COMA in vitro through the Ctf19 C-terminus whose deletion affected chromosome segregation fidelity in Sli15 wild-type cells. Tethering Sli15 to Ame1/Okp1 rescued synthetic lethality upon Ctf19 depletion in a Sli15 centromere-targeting deficient mutant. This study shows molecular characteristics of the point-centromere kinetochore architecture and suggests a role for the Ctf19 C-terminus in mediating CPC-binding and accurate chromosome segregation.

DOI: https://doi.org/10.7554/eLife.42879.001

## Introduction

Kinetochores enable the precise distribution of chromosomes during the eukaryotic cell division to avoid aneuploidy (*Santaguida and Musacchio, 2009*) which is associated with tumorigenesis, congenital trisomies and aging (*Baker et al., 2005*; *Pfau and Amon, 2012*). Faithful segregation of the duplicated sister chromatids relies on their exclusive attachment to spindle microtubules emerging from opposite spindle poles (*Foley and Kapoor, 2013*). The physical link between chromosomal DNA and microtubules is the kinetochore, a macromolecular protein complex that mediates the processive binding to depolymerizing microtubules driving the sister chromatids apart into the two emerging cells (*Biggins, 2013*; *Musacchio and Desai, 2017*). Kinetochore assembly is restricted to centromeres, chromosomal domains that are marked by the presence of the histone H3 variant Cse4[CENP-A] (human ortholog names are superscripted if appropriate) (*Earnshaw and Rothfield,*

*1985*; *Fukagawa and Earnshaw, 2014*). In humans, regional centromeres span megabases of DNA embedding up to 200 CENP-A containing nucleosomal core particles (NCPs) (*Bodor et al., 2014*; *Musacchio and Desai, 2017*). In contrast, *Saccharomyces cerevisiae* has point centromeres, which are characterized by a specific ~125 bp DNA sequence wrapped around a single Cse4-containing histone octamer (*Fitzgerald-Hayes et al., 1982*; *Camahort et al., 2009*; *Hasson et al., 2013*).

The budding yeast kinetochore is composed of about 45 core subunits which are organized in different stable complexes (*De Wulf et al., 2003*; *Westermann et al., 2003*) of which several are present in multiple copies (*Joglekar et al., 2006*). The kinetochore proteins are evolutionary largely conserved between yeast and humans (*Westermann and Schleiffer, 2013*; *van Hooff et al., 2017*) and share a similar hierarchy of assembly from DNA to the microtubule binding interface (*De Wulf et al., 2003*). The centromere proximal region is established by proteins of the Constitutive Centromere Associated Network (CCAN), also known as the CTF19 complex (CTF19c) in budding yeast. The CTF19c comprises the Chl4/Iml3$^{CENP-N/L}$, Mcm16/Ctf3/Mcm22$^{CENP-H/I/K}$, Cnn1/Wip1$^{CENP-T/W}$, Mhf1/Mhf2$^{CENP-S/X}$ and Ctf19/Okp1/Mcm21/Ame1$^{CENP-P/Q/O/U}$ (COMA) complexes plus Mif2$^{CENP-C}$ (*Cheeseman et al., 2002*; *Westermann et al., 2003*; *Biggins, 2013*; *Musacchio and Desai, 2017*) and the budding-yeast specific Nkp1/Nkp2 heterodimer. Another yeast inner kinetochore complex, the CBF3 (Ndc10/Cep3/Ctf13/Skp1) complex, has been identified as sequence-specfic binder of the centromeric DNA sequence CDEIII (*Ng and Carbon, 1987*; *Lechner and Carbon, 1991*). The CTF19c$^{CCAN}$ provides a cooperative high-affinity binding environment for the Cse4$^{CENP-A}$-NCP (*Weir et al., 2016*), where distinct subunits selectively recognize Cse4$^{CENP-A}$ specific features. Across different species the CENP-C signature motif interacts with divergent hydrophobic residues of the CENP-A C-terminal tail (*Musacchio and Desai, 2017*). Electron microscopy studies have recently resolved the interaction of CENP-N with the CENP-A centromere-targeting domain (CATD) in vertebrates (*Carroll et al., 2009*; *Guse et al., 2011*; *Pentakota et al., 2017*; *Chittori et al., 2018*; *Tian et al., 2018*). For budding yeast Cse4, a direct interaction has so far only been demonstrated with Mif2 (*Westermann et al., 2003*; *Xiao et al., 2017*). Apart from Mif2, the only essential CTF19c$^{CCAN}$ proteins are Ame1 and Okp1 (*Meluh and Koshland, 1997*; *Ortiz et al., 1999*; *De Wulf et al., 2003*), with the N-terminus of Ame1 binding the N-terminal domain of Mtw1 and thus serving as docking site for the outer kinetochore KMN network (KNL1$^{SPC105}$-/MIS12$^{MTW1}$-/NDC80$^{NDC80}$-complexes) (*Hornung et al., 2014*; *Dimitrova et al., 2016*).

The kinetochore is also a hub for feedback control mechanisms that ensure high fidelity of sister chromatid separation by relaying the microtubule attachment state to cell cycle progression, known as spindle assembly checkpoint (SAC), and by destabilizing improper kinetochore-microtubule attachments and selectively stabilizing the correct bipolar attachments, referred to as error correction mechanism (*Foley and Kapoor, 2013*; *Krenn and Musacchio, 2015*). A major effector of both regulatory feedback loops is the kinase Ipl1$^{Aurora\ B}$, a subunit of the evolutionary conserved tetrameric chromosomal passenger complex (CPC) which associates close to the centromere from G1 until anaphase (*Biggins and Murray, 2001*; *Widlund et al., 2006*; *Carmena et al., 2012*). The kinase subunit Ipl1$^{Aurora\ B}$ binds to the C-terminal IN-box domain (*Adams et al., 2000*; *Kaitna et al., 2000*) of the scaffold protein Sli15$^{INCENP}$, and Nbl1$^{Borealin}$ and Bir1$^{Survivin}$ form a three-helix bundle with the Sli15 N-terminus (*Klein et al., 2006*; *Jeyaprakash et al., 2007*). All known mechanisms for recruitment of the CPC to the yeast centromere rely on Bir1, which directly associates with Ndc10 (*Cho and Harrison, 2011*) and is recruited through Sgo1 to histone H2A phosphorylated at S121 by Bub1 which so far has only been established in fission yeast (*Kawashima et al., 2010*). Based on previous reports we refer to the CPC recruited through Ndc10 or H2A-P as centromere-targeted CPC pool, notwithstanding that the centromere-targeted Sli15$^{INCENP}$ scaffold may extend to, and Ipl1$^{Aurora\ B}$ may operate at, the kinetochore structure. CPC lacking the centromere-targeting domain (CEN) of Sli15$^{INCENP}$ is indicated as inner kinetochore-localized CPC (*Knockleby and Vogel, 2009*; *Musacchio and Desai, 2017*).

During early mitosis incorrect microtubule attachment states are resolved by Ipl1$^{Aurora\ B}$ which phosphorylates Ndc80 and Dam1 sites within the microtubule binding interface and thereby reduces their affinity towards microtubules (*Cheeseman et al., 2002*; *Miranda et al., 2005*; *Westermann et al., 2005*; *Cheeseman et al., 2006*; *DeLuca et al., 2006*; *Santaguida and Musacchio, 2009*). The selective destabilization promotes the establishment of a correctly bi-oriented kinetochore configuration at the mitotic spindle, referred to as amphitelic attachment (*Tanaka et al., 2002*). The spatial separation model for establishing biorientation (*Krenn and Musacchio, 2015*)

implies that centromere-targeting of Sli15 allows substrate phosphorylation by Ipl1[Aurora B] within the span of the Sli15[INCENP] scaffold and that tension dependent intra-kinetochore stretching (*Joglekar et al., 2009*) pulls the microtubule binding interface out of reach of Ipl1[Aurora B] resulting in dephosphorylation of outer kinetochore substrates and stabilization of amphitelic kinetochore-micro-tubule attachments (*Liu et al., 2009*; *Lampson and Cheeseman, 2011*).

A recent study challenged this model by showing that a Sli15 mutant lacking the centromere-targeting domain, Sli15ΔN2-228 (Sli15ΔN), suppressed the deletion phenotypes of Bir1, Nbl1, Bub1 and Sgo1 that mediate recruitment of the CPC to the centromere (*Campbell and Desai, 2013*). In contrast to wild-type Sli15, which localized between sister kinetochore clusters, Sli15ΔN showed weak localization overlapping with Nuf2 at kinetochores (*Campbell and Desai, 2013*). Apart from the altered localization, Sli15ΔN was indistinguishably viable from wild-type and displayed no significant chromosome segregation defects (*Campbell and Desai, 2013*; *Hengeveld et al., 2017*). Similarly, a survivin mutant in chicken DT40 cells that failed to localize INCENP and Aurora B to centromeres from prophase to metaphase displayed normal growth kinetics (*Yue et al., 2008*). These findings suggest that centromere-targeting of Sli15/Ipl1 is largely dispensable for error correction and SAC signaling. But a molecular understanding of how the inner kinetochore-localized Sli15ΔN/Ipl1 retains its biological function is missing.

We describe here the use of chemical crosslinking and mass spectrometry (XLMS) (*Herzog et al., 2012*) together with biochemical reconstitution to characterize the CTF19c[CCAN] subunit connectivity and the protein interfaces that establish a selective Cse4-NCP binding environment. Subunits of the COMA complex were previously implicated in CPC function at kinetochores (*De Wulf et al., 2003*; *Knockleby and Vogel, 2009*) and the Sli15ΔN mutant showed synthetic lethality with deletions of Ctf19 or Mcm21 (*Campbell and Desai, 2013*). Thus, we investigated whether the COMA complex directly associates with Sli15/Ipl1. We demonstrate that the Cse4-N-terminus (*Chen et al., 2000*) binds Ame1/Okp1 through the Okp1 core domain (*Schmitzberger et al., 2017*) and that dual recognition of budding yeast Cse4-NCP is established through selective interactions of the essential CTF19c[CCAN] proteins Mif2 and Ame1/Okp1 with distinct Cse4 motifs. We further show that Sli15/Ipl1 interacts with the Ctf19 C-terminus and that synthetic lethality upon Ctf19 depletion in the *sli15ΔN* background is rescued by fusing Sli15ΔN to the COMA complex. Our findings show contacts important for CTF19c[CCAN] architecture assembled at budding yeast point centromeres and indicate that the interaction of CPC and COMA is important for faithful chromosome segregation.

## Results

### The Ame1/Okp1 heterodimer selectively binds Cse4 containing nucleosomes

To screen for direct interaction partners of Cse4-NCPs we reconstituted the individual CTF19c[CCAN] subcomplexes (Mif2, Ame1/Okp1, Ctf19/Mcm21, Chl4/Iml3, Mcm16/Ctf3/Mcm22, Cnn1/Wip1, Nkp1/Nkp2, Mhf1/Mhf2) with Cse4- or H3-NCPs in vitro. The CTF19c[CCAN] complexes were purified either from bacteria or insect cells as homogenous and nearly stoichiometric complexes (*Figure 1B*). Consistent with a recent study (*Xiao et al., 2017*), using electrophoretic mobility shift assays (EMSA), we observed that Mif2 selectively interacted with Cse4-NCPs and not with H3-NCPs (*Figure 1A*). We also found that Ame1/Okp1 bound specifically to Cse4-NCPs (*Figure 1A*). The lack of interaction with H3-NCPs, which were reconstituted using the same 601 DNA sequence (*Tachiwana et al., 2011*), suggests that Ame1/Okp1 directly and selectively binds Cse4 and that the interaction does not require AT-rich DNA sequences as previously proposed (*Hornung et al., 2014*). In contrast to the EMSA titration of human CCAN complexes with CENP-A-NCP (*Weir et al., 2016*) using 10 nM NCP mixed with up to 20-fold excess of the respective subcomplexes, we could not detect Cse4-NCP band shifts with Chl4/Iml3, the orthologs of human CENP-NL, and with Mcm16/Ctf3/Mcm22, the orthologs of human CENP-HIK (no *S. cerevisiae* ortholog of CENP-M has been identified) using 500 nM NCP incubated with a twofold excess of the complexes. Ctf19/Mcm21, Cnn1/Wip1, Nkp1/Nkp2 and Mhf1/Mhf2 did also not form distinct complexes with either Cse4- or H3-NCPs in the EMSA indicating that Mif2 and Ame1/Okp1 possess a higher relative binding affinity to Cse4-NCPs than the other CTF19c subcomplexes (*Figure 1A*).

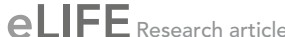

**Figure 1.** The heterodimeric Ame1/Okp1 complex directly and selectively binds the Cse4-NCP. (**A**) Electrophoretic mobility shift assays (EMSAs) of the indicated CTF19c[CCAN] subunits and subcomplexes mixed in a 2:1 molar ratio with either Cse4- or H3-NCPs. DNA/protein complexes were separated on a 6% native polyacrylamide gel. The DNA is visualized by SYBR Gold staining. (**B**) Coomassie stained gel of the individual inner kinetochore components, recombinantly purified from *E. coli*, used in the EMSA in (**A**). (**C**) XLMS analysis of the in vitro reconstituted Cse4-NCP:Mif2:COMA:Chl4/Iml3:MTW1c complex. Proteins are represented as bars indicating annotated domains (*Supplementary file 3*) according to the color scheme in the legend. Subunits of a complex are represented in the same color and protein lengths and cross-link sites are scaled to the amino acid sequence.

DOI: https://doi.org/10.7554/eLife.42879.002

The following figure supplement is available for figure 1:

**Figure supplement 1.** Size exclusion chromatography (SEC) of the in vitro reconstituted Ctf19/Mcm21/Ame1/Okp1 (COMA):Chl4/Iml3:Mif2:MTW1c: Cse4-NCP complex.

DOI: https://doi.org/10.7554/eLife.42879.003

To identify the binding interfaces of the Ame1/Okp1:Cse4-NCP complex we performed XLMS analysis. We reconstituted a complex on Cse4-NCP composed of Ame1/Okp1, Mif2, Ctf19/Mcm21, Chl4/Iml3 and the MTW1c which links the KMN network to the inner kinetochore receptors Ame1 and Mif2 (*Przewloka et al., 2011*; *Screpanti et al., 2011*; *Hornung et al., 2014*). Size-exclusion chromatography (SEC) analysis showed that MTW1c forms a complex with Ame1/Okp1, Mif2, Ctf19/Mcm21 and Chl4/Iml3 and the peak fraction shifted to a higher molecular weight upon addition of Cse4-NCPs depicting nearly stoichiometric protein levels of all subunits (*Figure 1—figure supplement 1*). In all in vitro reconstitution and XLMS experiments we used wild-type MTW1c lacking the phosphorylation mimicking mutations of Dsn1 S240 and S250, which have been shown to stabilize the interaction with Mif2[CENP-C] and Ame1[CENP-U] (*Akiyoshi et al., 2013*; *Dimitrova et al., 2016*), but were not required for complex formation on SEC columns (*Figure 2C*, *Figure 1—figure supplement*

**Figure 2.** A short helical motif within the Cse4 N-terminus serves as Ame1/Okp1 docking site and is essential in vivo. (**A**) Multiple sequence alignment of Cse4^CENP-A proteins. Yeast protein sequences with the highest similarities to *S. cerevisiae* Cse4, three mammalian and the *S. pombe* homologous CENP-A protein sequences were included in the alignment. The amino acid (aa) patch, conserved in interrelated yeasts, is highlighted in pink (*S. cerevisiae* Cse4 aa 34–61). The RG motif in the mammalian sequences is indicated by arrowheads. Amino acid residues are colored and annotated

*Figure 2 continued on next page*

*Figure 2 continued*
according to the ClustalW color and annotation codes (S.: *Schizosaccharomyces*, C.: *Candida*, Z.: *Zygosaccharomyces*, L.: *Lachancea*). Residues that are identical among aligned protein sequences (*), conserved substitutions (:), and semiconserved substitutions (.) are indicated. (B) Scheme of the deletion mutants within the Cse4 N-terminus used in the SEC experiments in (C) and (D) and in the cell viability assays in (E). The conserved region (aa 34–61) is highlighted in pink. (C) SEC analysis of the indicated mixtures of recombinant Ame1/Okp1 (AO) and MTW1c and reconstituted H3-, Cse4-, Cse4Δ2–30- or Cse4Δ31–60-NCPs. Ame1/Okp1, MTW1c and the Cse4 proteins were mixed equimolar. Eluted proteins were visualized by SDS-PAGE and Coomassie staining. (D) SEC analysis of Ame1/Okp1 (AO) preincubated with Cse4Δ34–46- or Cse4Δ48–61-NCPs. Eluted complexes were analyzed by SDS-PAGE and Coomassie staining. (E) Left panel: Cell growth assay of Cse4 mutants in budding yeast using the anchor-away system. The Cse4 wild-type and indicated mutant proteins were ectopically expressed in a Cse4 anchor-away strain (Cse4-FRB) and cell growth was monitored by plating 1:10 serial dilutions on YPD medium at 30°C in the absence or presence of 1 µg/ml rapamycin. Right panel: Western blot analysis of the ectopically expressed Cse4 wild-type and mutant protein levels in the yeast strains shown on the left. Pgk1 levels are shown as loading control.
DOI: https://doi.org/10.7554/eLife.42879.004

1). In total 349 inter-subunit crosslinks between the fifteen proteins were identified (*Figure 1C*, *Supplementary file 1*). The majority of the crosslinks detected within the different subcomplexes MTW1c, COMA, Chl4/Iml3, and Cse4-NCP are in agreement with previous studies validating our crosslink map (*De Wulf et al., 2003*; *Hinshaw and Harrison, 2013*; *Hornung et al., 2014*). Moreover, crosslinks from the Mif2 N-terminus to the MTW1c (*Przewloka et al., 2011*; *Screpanti et al., 2011*), from the Mif2 Chl4/Iml3-binding domain to Chl4 (*Hinshaw and Harrison, 2013*), and from the Mif2 signature motif to the Cse4 C-terminus (*Figure 1C*, *Supplementary file 1*) (*Kato et al., 2013*) are consistent with previously described interfaces. Crosslinks between Ame1/Okp1 and Cse4 occur exclusively between Okp1 and Cse4, suggesting that Okp1 is the direct binding partner of Cse4. Furthermore, Okp1 was the only COMA subunit that crosslinked to the three canonical histones H2A, H2B and H4 with the exception of one crosslink between Ame1 and H2A. Our analysis indicated a close association between Chl4/Iml3 and all COMA subunits. A direct interaction between COMA and Chl4 was reported previously and the Ctf19/Mcm21 heterodimer was found to be required for the kinetochore localization of Chl4 and Iml3 (*Schmitzberger et al., 2017*).

## The essential N-terminal domain of Cse4 is required for Okp1 binding

To further characterize the interaction between Ame1/Okp1 and Cse4-NCPs we aimed to identify the binding interface of the Ame1/Okp1:Cse4-NCP complex. Two crosslinks were detected between Okp1 and the essential Cse4 N-terminus (*Figure 1C*, *Supplementary file 1*). A multiple sequence alignment (MSA) of Cse4^CENP-A protein sequences (*Figure 2A*) detected a conserved region (ScCse4 aa 34–61), unique to Cse4 proteins of interrelated yeasts, which is almost identical to the so-called 'essential N-terminal domain' (END), aa 28–60, shown to be required for the essential function of the Cse4 N-terminus and for recruiting the 'Mcm21p/Ctf19p/Okp1p complex' to minichromosomes (*Keith et al., 1999*; *Ortiz et al., 1999*; *Chen et al., 2000*).

To assess whether the Cse4 END mediates the interaction with Ame1/Okp1 we tested binding of recombinant Ame1/Okp1 to reconstituted wild-type and deletion mutants (*Figure 2B*) of Cse4- and to H3-NCPs by SEC (*Figure 2C*). Wild-type Cse4-NCP but not H3-NCP formed a stoichiometric complex with Ame1/Okp1 (*Figure 2C*) which is consistent with our EMSA and XLMS analyses (*Figure 1A,C*). In addition, Ame1/Okp1 bound to a Cse4-NCP retained the ability to interact with the MTW1c (*Hornung et al., 2014*), forming a direct link between the KMN network and the centromeric nucleosome (*Figure 2C*). Truncation of the first 30 N-terminal residues of Cse4 neither affected its ability to bind Ame1/Okp1, nor was it essential for viability (*Figure 2C*) (*Chen et al., 2000*). However, the Cse4Δ31–60 mutant abrogated Ame1/Okp1:Cse4-NCP complex formation (*Figure 2C*). To further narrow down the interface, two deletion mutants splitting the END in half, Cse4Δ34–46 and Cse4Δ48–61 (*Figure 2B*), were tested in SEC experiments. While Cse4Δ48–61 associated with Ame1/Okp1, deletion of amino acids 34–46 completely disrupted the interaction (*Figure 2D*). All Cse4 N-terminal mutant and wild-type NCPs eluted at similar retention times from the SEC column indicating that the Cse4 N-terminal deletions did not affect Cse4 incorporation and stability of the nucleosomes (*Figure 2C,D*).

The crosslink-derived distance restraints as well as SEC analysis identified a conserved Cse4 peptide motif of amino acids 34–46 which is necessary for Ame1/Okp1 interaction. To test whether this motif is essential for cell viability, we depleted endogenous Cse4 from the nucleus using the anchor-

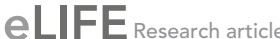 Research article

Biochemistry and Chemical Biology | Cell Biology

away technique and performed rescue experiments by ectopically expressing the Cse4 mutants Cse4Δ34–46 and Cse4Δ48–61. Indeed, deletion of amino acids 34–46 was lethal, whereas the Cse4Δ48–61 mutant displayed wild-type growth rates (*Figure 2E*). The observation that deletion of the minimal Ame1/Okp1 interacting Cse4 motif (aa 34–46) correlates with the loss of cell viability, whereas the C-terminal half of the END (aa 48–61) is neither essential for viability nor required for Ame1/Okp1 association suggests that binding of the Ame1/Okp1 heterodimer to Cse4 residues 34–46 is essential for yeast growth. The Mif2 signature motif (*Xiao et al., 2017*) and Ame1/Okp1 recognize distinct motifs at the Cse4 C- and N-terminus (*Figure 1C*), respectively, and both are essential for viability (*Hornung et al., 2014*).

## The Okp1 core domain interacts with Cse4

To characterize the Cse4 binding site in Okp1 we applied crosslink-derived restraints to narrow down the putative interface to amino acids 95–202 of Okp1 (*Figure 1C*, *Supplementary file 1*). Based on MSA analysis of Okp1 sequences, this region harbors a conserved stretch (aa 127–184), including part of the previously described Okp1 core domain (aa 166–211) which is essential for cell growth and whose function is still elusive (*Schmitzberger et al., 2017*) (*Figure 3A*). Furthermore, a secondary structure analysis predicted two alpha helices within the conserved domain (helix1 aa 130–140, helix2 aa 156–188) (*Figure 3A*). Thus, we designed three deletion mutants (Okp1Δ123–147, Okp1Δ140–170, Okp1Δ163–187) and purified all Okp1 mutant proteins in complex with Ame1 from *E. coli*. In EMSAs Ame1/Okp1Δ123–147 bound to Cse4-NCPs as well as did the wild-type Ame1/Okp1 complex, whereas Ame1/Okp1Δ140–170 associated only weakly and Ame1/Okp1Δ163–187 failed to associate with Cse4-NCPs (*Figure 3B*). These results are consistent with monitoring protein complex formation by SEC (*Figure 3—figure supplement 1*). In addition, analysis of the Okp1 deletion mutants Δ123–147 and Δ163–187 in cell viability assays showed a tight correlation between their requirement for the interaction with Cse4 and being essential for yeast growth (*Figure 3C*) (*Schmitzberger et al., 2017*). This finding further supports the notion that the recognition of the Cse4 nucleosome by Ame1/Okp1 is essential in budding yeast.

## The COMA complex interacts with Sli15/Ipl1 through the Ctf19 C-terminus

The COMA complex is composed of two essential, Ame1/Okp1, and two non-essential, Ctf19/Mcm21, subunits (*Ortiz et al., 1999*; *Cheeseman et al., 2002*). Both, Ctf19 and Mcm21 contain C-terminal tandem-RWD (RING finger and WD repeat containing proteins and DEAD-like helicases) domains forming a rigid heterodimeric Y-shaped scaffold whose respective N-terminal RWDs of the tandems pack together as shown by a crystal structure of the *K. lactis* complex (*Schmitzberger and Harrison, 2012*). The *ctf19Δ* or *mcm21Δ* mutants become synthetically lethal in a *sli15ΔN* background (*Campbell and Desai, 2013*). Furthermore, Ame1 has been suggested to have a role in Sli15 localization close to kinetochores independently of Bir1 (*Knockleby and Vogel, 2009*). To investigate whether Sli15/Ipl1 associates with the COMA complex, in vitro reconstitution and XLMS analysis detected 98 inter-protein and 69 intra-protein crosslinks (*Figure 4A*, *Supplementary file 2*). In particular, there were 10 crosslinks from the C-terminal RWD (RWD-C) domain of Ctf19 and 4 crosslinks from the Mcm21 RWD-C domain to the microtubule binding domain of Sli15 (aa 229–565) (*Figure 4A*, *Supplementary file 2*, *3*). In the Ame1/Okp1 heterodimer, we identified crosslinks from Sli15 to Okp1 and from Ipl1 to Ame1. The crosslink detected to lysine 366 of Okp1 is located near the identified Ctf19/Mcm21 binding site within Okp1 ('segment 1' aa 321–329) (*Schmitzberger et al., 2017*) and thus is close to the RWD-C domains of Ctf19 and Mcm21. We verified the interaction of Sli15 and the Ctf19 RWD-C domain by in vitro binding assays. Sli15-2xStrep/Ipl1 was immobilized on Streptavidin beads and incubated with a 2-fold molar excess of either Ame1/Okp1 and Ctf19/Mcm21 using wild-type Ctf19 protein or a C-terminal deletion mutant Ctf19Δ270–369 (Ctf19ΔC). Ame1/Okp1 and Ctf19/Mcm21 were both pulled down with Sli15/Ipl1 either as individual complexes or in combination (*Figure 4B*). In agreement with previous findings (*Schmitzberger et al., 2017*), recombinant Ctf19ΔC formed a stoichiometric complex with Mcm21, but lost its ability to bind Sli15/Ipl1 indicating that the RWD-C of Ctf19 is required for Sli15/Ipl1 interaction in vitro (*Figure 4C*). Autophosphorylation of Sli15/Ipl1 abrogated its interaction with Ame1/Okp1 and Ctf19/Mcm21 indicating that like the phosphorylation-regulated binding to



**Figure 3.** The essential core domain of Okp1 is required for the interaction with Cse4-NCPs. (**A**) Multiple sequence alignment of Okp1 amino acid sequences from related yeast species. Amino acid residues of the conserved region are colored and annotated according to the ClustalW color and annotation codes. Green bars above the alignment represent alpha helical regions predicted by Jpred (*Drozdetskiy et al., 2015*). Lines below the alignment indicate the overlapping Okp1 deletion mutants analysed in (**B**) and (**C**). Residues that are identical among aligned protein sequences (*), conserved substitutions (:), and semiconserved substitutions (.) are indicated. (**B**) EMSA assessing complex formation of Cse4-NCPs with Ame1/ Okp1 including wild-type (wt) Okp1 and the indicated Okp1 deletion mutants. Recombinant Ame1/Okp1 complexes were tested in a 1:1 (1) and 2:1 (2) molar ratio with Cse4-NCPs. The DNA is visualized by SYBR Gold staining. (**C**) Cell viability assay of Okp1 deletion mutants using the anchor away (aa) technique. Yeast growth of either the untransformed (-) Okp1 anchor-away strain (Okp1-FRB) or of strains transformed with the indicated Okp1 rescue alleles was tested in 1:10 serial dilutions on YPD medium in the absence or presence of 1 µg/ml rapamycin for 72 hr at 30℃.

*Figure 3 continued on next page*

*Figure 3 continued*

DOI: https://doi.org/10.7554/eLife.42879.005

The following figure supplement is available for figure 3:

**Figure supplement 1.** Identification of the Cse4 binding site on Okp1.

DOI: https://doi.org/10.7554/eLife.42879.006

microtubules, phosphorylation of Sli15 by Ipl1 may prevent and regulate its binding to the COMA complex (*Figure 4B*).

In summary, crosslink-derived restraints identified the Ctf19 RWD-C domain as a Sli15/Ipl1 docking site within the COMA complex, a conclusion supported by the loss of interaction upon deletion of the Ctf19 C-terminus in vitro.

## Tethering Sli15ΔN selectively to COMA rescues the synthetic lethality of a *sli15ΔN* mutant upon Ctf19 depletion

As deletions of Ctf19 or Mcm21 were synthetically lethal in a *sli15ΔN* background (*Campbell and Desai, 2013*) and Sli15 associated with the Ctf19 RWD-C in vitro (*Figure 4*), we investigated the relevance of this interaction by performing yeast viability assays. First, we reproduced the reported synthetic lethality by anchoring-away Ctf19-FRB in a yeast strain, in which the endogenous *SLI15* copy was replaced by *sli15ΔN* (*Haruki et al., 2008*). We found that in the presence of Ctf19-FRB, cells expressing Sli15ΔN are viable, but display synthetic lethality on rapamycin containing medium, consistent with previous findings (*Campbell and Desai, 2013*) (*Figure 5A*).

Recently, the Ctf19 N-terminus has been identified as the receptor domain of the cohesin loading complex Scc2/4 in late G1 phase (*Hinshaw et al., 2017*). To address whether Sli15/Ipl1 has an active role in this process, we deleted 30 amino acids of Ctf19 (Ctf19ΔN2-30) which have been shown to contain phosphorylation sites of the Dbf4-dependent kinase required for Scc2/4 recruitment to the centromere (*Hinshaw et al., 2017*). Cells expressing Ctf19ΔN2-30 in the *sli15ΔN* background were just as viable upon depletion of Ctf19-FRB as those expressing intact Ctf19 (*Figure 5A*), demonstrating that the synthetic lethality is independent of the Ctf19 N-terminus and its role in cohesin loading.

If the synthetic effect is associated with the loss of interaction between Sli15ΔN and COMA, artificial tethering of Sli15ΔN to the kinetochore should restore growth. We generated fusions of Sli15ΔN to various inner and outer kinetochore proteins and investigated whether growth was restored in a *CTF19-FRB/sli15ΔN* background. Ectopic expression of Sli15ΔN fusions to the outer kinetochore subunits Mtw1 or Dsn1 and to the inner kinetochore subunits Mif2, Ctf3 or Cnn1 did not rescue viability (*Figure 5B*). But selectively tethering Sli15ΔN to Ame1 or Okp1 restored growth (*Figure 5B*).

We further tested whether the rescue of synthetic lethality depended on the Sli15 single alpha helix domain (SAH, aa 516–575) (*Samejima et al., 2015*; *van der Horst et al., 2015*; *Fink et al., 2017*) and the Ipl1 binding domain (IN-box, aa 626–698) (*Adams et al., 2000*; *Kang et al., 2001*). Both domains are essential for cell growth in the Sli15 wild-type or the *sli15ΔN* background (*Figure 5C*) (*Kang et al., 2001*). Cells ectopically expressing the Sli15ΔN mutant protein grew like wild-type, but displayed sensitivity to 15 μg/ml benomyl which contrasted the previous observation that cells carrying the endogenous *sli15ΔN* allele were not sensitive to 12.5 μg/ml benomyl (*Campbell and Desai, 2013*). These deviating observations may be a result of different experimental conditions. To distinguish the requirement of one domain from that of the other in the context of inner kinetochore-localized Sli15/Ipl1, we generated Ame1- and Okp1-Sli15ΔN fusion constructs in which either the IN-box or the SAH domain of Sli15ΔN had been deleted. While expression of Ame1- or Okp1-Sli15ΔNΔSAH proteins rescued cell growth in the *sli15ΔN* background upon Ctf19 depletion, Ame1- and Okp1-Sli15ΔNΔIN fusions did not, indicating that Ipl1 kinase activity is required (*Figure 5D*). Since the ectopically expressed fusion proteins were tested in the *sli15ΔN* background, the result indicates that Ipl1 activity associated with endogenous Sli15ΔN could not rescue synthetic lethality and that tethering Ipl1 activity to COMA subunits is crucial. In contrast, deletion of the SAH domain in Ame1- and Okp1-Sli15ΔNΔSAH fusions was not lethal and was presumably rescued by the SAH domain of the endogenous Sli15ΔN protein (*Figure 5D*) suggesting that the SAH domain is not required for the function of the inner kinetochore-localized CPC pool.



**Figure 4.** The core-CPC Sli15/Ipl1 associates with the COMA complex through the Ctf19 C-terminal RWD domain in vitro. (**A**) Network representation of lysine-lysine cross-links identified on recombinant Sli15/Ipl1 in complex with COMA. Proteins are represented as bars indicating annotated domains (*Supplementary file 3*) according to the color scheme in the legend. Subunits of a complex are represented in the same color. Protein lengths and cross-link sites are scaled to the amino acid sequence. (**B**) In vitro binding assay analyzing the interaction of Sli15/Ipl1 with the COMA complex.
*Figure 4 continued on next page*

*Figure 4 continued*

Recombinant Sli15-2xStrep/Ipl1 was immobilized on Streptavidin beads and incubated with Ctf19/Mcm21, Ame1/Okp1 or Ame1/Okp1/Ctf19/Mcm21. Autophosphorylation (p) of Sli15/Ipl1 largely reduced bound protein levels. Dephosphorylation (dp) of Sli15/Ipl1 did not alter the bound proteins levels, which were visualized by SDS-PAGE and Coomassie staining. (C) In vitro binding assay analyzing the interaction of Sli15/Ipl1 with Ctf19/Mcm21 or Ctf19ΔC/Mcm21. Ctf19ΔC lacks the last 100 amino acids which form the C-terminal RWD domain. This panel is representative of three independent experiments.

DOI: https://doi.org/10.7554/eLife.42879.007

### Ame1- or Okp1-Ctf19 fusion proteins require the Ctf19 RWD-C domain to rescue synthetic lethality of a *sli15ΔN* mutant strain upon Ctf19 depletion

Since the RWD-C domain of Ctf19 was required for association with Sli15/Ipl1 in vitro (*Figure 4C*), we asked whether its deletion would cause synthetic lethality with *sli15ΔN*. As recently described, the Ctf19 C-terminus is involved in formation of the COMA complex through binding to Okp1 (*Schmitzberger et al., 2017*) and consequently, its deletion abrogates kinetochore localization of Ctf19 (*Figure 6—figure supplement 1*). To circumvent loss of Ctf19 from kinetochores, we tested whether Ame1 or Okp1 fusions to wild-type Ctf19 or Ctf19ΔC were able to rescue synthetic lethality in the *sli15ΔN/CTF19-FRB* background. Fusions to both, the N- or C-terminus, of wild-type Ctf19 restored viability, whereas fusions to Ctf19ΔC resulted in synthetic lethality (*Figure 6A*) suggesting that recruitment of Ipl1 activity to the inner kinetochore mediated by the Ctf19 C-terminus is important for CPC function.

### Deletion of the Ctf19 RWD-C domain causes a chromosome segregation defect in the Sli15 wild-type background

Since Ctf19 mutants display normal growth, but have chromosome segregation defects (*Hyland et al., 1999*), we tested whether the Ctf19 C-terminus is important for this function using the minichromosome loss assay (*Hieter et al., 1985*). The Ctf19 anchor-away strain was transformed simultaneously with the various Ctf19 rescue constructs and a centromeric plasmid carrying the *SUP11* gene as a marker which indicated loss of the minichromosome by red pigmentation (*Hieter et al., 1985*). Depletion of Ctf19 from the nucleus resulted in a severe chromosome segregation defect that was not observed by growing cells on medium lacking rapamycin which showed 4% red/sectored colonies (*Figure 6B*). Ectopic expression of the Ctf19 wild-type protein decreased the segregation defect to 19% red/sectored colonies (*Figure 6B*, *Figure 6B—source data 1*) and fusion of Okp1 to the C-terminus of wild-type Ctf19 reduced the red/sectored colonies to 32%. But the fusion of Okp1 to the Ctf19 N-terminus (Okp1-Ctf19 and Okp1-Ctf19ΔC) did not rescue the segregation defect (*Figure 6—figure supplement 2*, *Figure 6B—source data 1*), indicating that the function of the Ctf19 N-terminus is compromised by fusing it to Okp1 (*Figure 6B*, *Figure 6B—source data 1*). Thus, the Ctf19-Okp1 fusion rescued the segregation defect, albeit to a slightly lesser extent than the Ctf19 wild-type protein. In contrast, Ctf19ΔC-Okp1, which was localized at the kinetochore (*Figure 6C*), was unable to rescue the segregation defect (*Figure 6B*, *Figure 6B—source data 1*) suggesting that the Ctf19 C-terminus has a role in mediating accurate chromosome segregation.

## Discussion

### The Ame1/Okp1 heterodimer directly links Cse4 nucleosomes to the outer kinetochore

We investigated the subunit connectivity of the inner kinetochore assembled at budding yeast point centromeres at the domain level using in vitro reconstitution and XLMS. We found that in addition to Mif2 (*Xiao et al., 2017*), the Ame1/Okp1 heterodimer of the COMA complex is a direct and selective interactor of Cse4-NCPs. We identified the conserved motifs aa 163–187 of the Okp1 core domain (*Figure 3B,C*) (*Schmitzberger et al., 2017*) and aa 34–46 (*Figure 2D,E*) of the Cse4 END to establish the interaction. Although, we did not address whether the Cse4 residues 34–46 are required for the Ame1/Okp1 kinetochore recruitment, the notion that the essential function of the

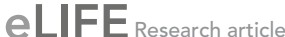

**Figure 5.** Synthetic lethality of Sli15ΔN and Ctf19 depletion is rescued by fusing Sli15ΔN to Ame1/Okp1 and is independent of Ctf19's role in cohesin loading. (**A**)-(**D**) Cell viability assays studying the rescue of synthetic lethality of a *sli15ΔN/CTF19-FRB* strain using the anchor-away system. The indicated constructs were transformed into a Ctf19 anchor-away (aa) strain (Ctf19-FRB) carrying *sli15ΔN* (ΔN) at the endogenous locus (A, B, D,) or into a Sli15 anchor-away strain (Sli15-FRB) (**C**). Yeast growth was tested in serial dilutions either untransformed (-) or transformed with the indicated rescue constructs on YPD medium in the absence or presence of 1 μg/ml rapamycin at 30°C. The lower panels in (**B**), (**C**) and (**D**) show western blot analysis of the ectopically expressed protein levels. Pgk1 levels are shown as loading control. (**A**) Deletion of the Ctf19 N-terminus (Ctf19ΔN2-30) does not affect cell viability in a *sli15ΔN* background. (**B**) Artificial tethering of Sli15ΔN to Ame1 or Okp1 rescued synthetic lethality of *sli15ΔN* cells upon Ctf19-FRB depletion from the nucleus. (**C**) Growth phenotypes of Sli15 wild-type, Sli15ΔSAH, Sli15ΔN, and Sli15ΔNΔSAH tested in a Sli15-FRB anchor-away strain. (**D**) Rescue of cell growth by ectopic Ame1-Sli15ΔN or Okp1-Sli15ΔN fusion proteins is dependent on the Sli15 Ipl1-binding domain (IN-box), whereas the SAH domain is dispensable.

DOI: https://doi.org/10.7554/eLife.42879.008



**Figure 6.** The Ctf19 C-terminus is important for chromosome segregation in the Sli15 wild-type background. (**A**) Left panel: Growth assay of the *sli15ΔN/CTF19-FRB* strain expressing Ame1-Ctf19, Ame1-Ctf19ΔC, Okp1-Ctf19, Okp1-Ctf19ΔC, Ctf19-Okp1 and Ctf19ΔC-Okp1 fusion proteins from the rescue plasmid. Right panel: Western blot analysis visualizing the levels of the ectopically expressed, C-terminally 7xFLAG-tagged fusion proteins. Pgk1 levels are shown as loading control. (aa: Anchor-away) (**B**) Minichromosome loss assay. Chromosome segregation fidelity was determined in the Ctf19

*Figure 6 continued on next page*

*Figure 6 continued*

anchor-away (*SLI15/CTF19-FRB*) strain, containing a minichromosome, either untransformed (-) or transformed with the indicated rescue constructs in the absence or presence of 1 µg/ml rapamycin. The percentage and standard error of red/red sectored colonies to the total colony number (white plus red/red sectored) of three biological replicates is shown. The results of 100% red colonies may be indicative of non-optimal conditions for the chromosome loss assay in combination with the anchor-away technique. (C) Localisation of ectopically expressed Ctf19-Okp1-GFP and Ctf19ΔC-Okp1-GFP fusion proteins in the Ctf19 anchor-away strain (*SLI15/CTF19-FRB*) in the presence of 1 µg/ml rapamycin. Live cell fluorescence microscopy was performed 3 hr after rapamycin addition. Ndc80-mCherry was used as kinetochore marker. Merged mCherry and GFP signals are shown on the right. (BF: brightfield).

DOI: https://doi.org/10.7554/eLife.42879.009

The following source data and figure supplements are available for figure 6:

**Source data 1.** Quantification of the minichromosome loss assay in a *SLI15/CTF19-FRB* strain.

DOI: https://doi.org/10.7554/eLife.42879.012

**Figure supplement 1.** Ctf19ΔC-GFP does not localize to kinetochores.

DOI: https://doi.org/10.7554/eLife.42879.010

**Figure supplement 2.** The N-terminal fusion protein of Ctf19 with Okp1 does not rescue chromosome segregation defects upon nuclear depletion of Ctf19 in the Sli15 wild-type background.

DOI: https://doi.org/10.7554/eLife.42879.011

Cse4 N-terminus and the binding interface for Ame1/Okp1 are mediated by the same 13 amino acid motif (*Figure 2*) suggests that Ame1/Okp1 is an essential link between centromeric nucleosomes and the outer kinetochore (*Hornung et al., 2014*).

Recent studies have identified the same Cse4 region to interact with Ame1/Okp1 (*Anedchenko et al., 2019*; *Hinshaw and Harrison, 2019*). Anedchenko et al. found that the affinity of Cse4 N-terminal peptides to Ame1/Okp1 increases with the peptide length up to the low nano-molar range and that methylation of Cse4 R37 and acetylation of Cse4 K49 significantly reduces the binding affinity. Similarly, this region is regulated by Ipl1 phosphorylation in vivo and phosphorylation-mimicking mutants have been found to suppress temperature-sensitive Ipl1 and phosphorylation-deficient Dam1 und Ndc80 mutations (*Boeckmann et al., 2013*), and to decrease the affinity of a Cse4 peptide to Ame1/Okp1 (*Hinshaw and Harrison, 2019*). This observation has interesting implications on the regulation of kinetochore assembly by Ipl1 destabilizing the Cse4-Ame1/Okp1 interaction in a cell cycle regulated manner. Moreover, weakening the interaction of Ame1/Okp1 with Cse4 may have a role in the tension sensing and error correction mechanisms (*Boeckmann et al., 2013*).

## Dual recognition of Cse4 at point centromeres by a CTF19c[CCAN] architecture distinct from vertebrate regional centromeres

In vertebrates, CENP-NL and CENP-C, interact directly and specifically with CENP-A. CENP-C binds divergent hydrophobic residues at the CENP-A C-terminus, whereas CENP-N associates with the CENP-A CATD (*Carroll et al., 2009*; *Carroll et al., 2010*; *Guse et al., 2011*; *Kato et al., 2013*; *Weir et al., 2016*; *Pentakota et al., 2017*). Recently, electron microscopy reconstructions of human CENP-A nucleosomes in complex with CENP-N/L identified the RG motif in the L1 loops of the CATD (*Zhou et al., 2011*) as the CENP-N interaction site in CENP-A (*Pentakota et al., 2017*; *Chittori et al., 2018*; *Tian et al., 2018*). We did not detect complex formation of Chl4/Iml3 with Cse4-NCPs in our EMSA (*Figure 1A*). Whether this observation can be attributed to the lack of conservation of the RG motif in the corresponding Cse4 sequences in related budding yeasts (*Figure 2A*), and whether this reflects a different role of Chl4/Iml3 in Cse4 recognition and kinetochore assembly remains to be determined. Our crosslink-derived restraints are also in good agreement with a recent cryo-electron microscopy structure of a 13-subunit budding yeast inner kinetochore complex lacking the Cse4-NCP and Mif2 (*Hinshaw and Harrison, 2019*) showing for instance crosslinks between the C-terminal domain of Chl4 and central regions of Ctf19 and Mcm21.

Similarly in humans, recruitment of the CENP-OPQRU complex to kinetochores requires a joint interface formed by CENP-HIKM and CENP-LN (*Foltz et al., 2006*; *Okada et al., 2006*; *Pesenti et al., 2018*), but loss of the complex does not affect localization of other inner kinetochore proteins. Differences between vertebrate and budding yeast inner kinetochores are reflected by the physiological importance of the involved proteins, as Ame1/Okp1 together with Mif2 are the

essential CTF19c[CCAN] proteins in budding yeast, whereas knockouts of CENP-U/Q in DT40 cells are viable (*Hori et al., 2008*).

## The Ctf19 C-terminus is required for Sli15/Ipl1 binding in vitro and has a role in accurate chromosome segregation

Although the Ctf19/Mcm21 heterodimer is not essential, *ctf19Δ* and *mcm21Δ* mutants have chromosome segregation and cohesion defects (*Hyland et al., 1999*; *Ortiz et al., 1999*; *Poddar et al., 1999*; *Fernius and Marston, 2009*; *Hinshaw et al., 2017*). Moreover, Ctf19 and Mcm21 become essential when centromere-targeting of the CPC is lost in a *sli15ΔN* mutant. This observation has led to the hypothesis that centromere-targeted Sli15 might be involved in cohesin loading or in cohesion maintenance (*Campbell and Desai, 2013*). An alternative model posits that COMA is required for the localization and positioning of Sli15/Ipl1 at the kinetochore (*Knockleby and Vogel, 2009*).

Our work showed that COMA interacts directly with Sli15/Ipl1 and identified the Ctf19 RWD-C domain as the primary docking site (*Figure 4A,C*). Synthetic lethality upon Ctf19 or Mcm21 depletion in a *sli15ΔN* background was rescued by fusions of Sli15ΔN to COMA subunits, whereas fusions to other inner or outer kinetochore proteins did not (*Figure 5B*). This observation suggests that positioning Sli15/Ipl1 proximal to Ame1/Okp1 is important in vivo. Because of the requirement of a functional Ipl1-binding IN-box on Sli15 for restoring viability we assume that the observed synthetic lethality is due to mislocalized Ipl1 kinase (*Figure 5D*). Tethering Sli15 to the inner kinetochore might ensure the spatial positioning of Ipl1 kinase activity towards outer kinetochore substrates (*Akiyoshi et al., 2013*; *Foley and Kapoor, 2013*; *Krenn and Musacchio, 2015*), required for correcting erroneous kinetochore-microtubule attachments (*Figure 7*). COMA-Sli15ΔN fusions lacking the SAH domain rescued growth, indicating that this domain is dispensable for CPC function at the inner kinetochore. Because the SAH domain is required for binding to spindle microtubules and critical for cell survival (*Samejima et al., 2015*; *van der Horst et al., 2015*; *Fink et al., 2017*), we infer that the observed rescue was mediated by the SAH domain of endogenous Sli15ΔN (*Figure 5D*).

We also showed that deletion of the Ctf19 RWD-C domain was sufficient to cause synthetic lethality with Sli15ΔN (*Figure 6A*) and that recombinant Ctf19ΔC in complex with Mcm21 (*Figure 4C*) does not interact with Sli15. Moreover, assessing the initially proposed model for the synthetic growth defect of Ctf19/Mcm21 deletion in a *sli15ΔN* background (*Campbell and Desai, 2013*), we observed that deletion of the Ctf19 N-terminus did not cause a synthetic effect in *sli15ΔN* mutant cells. This result indicated that the synthetic growth defect is mediated by a Ctf19 domain distinct from its N-terminus and its role in cohesin loading.

Apart from the synthetic condition we addressed whether the Ctf19 C-terminus is required for chromosome segregation in Sli15 wild-type cells by monitoring missegregation in a minichromosome loss assay (*Hieter et al., 1985*). We showed that loss of the centromeric plasmid upon Ctf19 depletion was rescued to 70% by the ectopic expression of Ctf19-Okp1 and this rescue was abrogated upon deletion of the Ctf19 RWD-C domain in the fusion protein (*Figure 6B*). Similar observations have been obtained in a concomitant study (*García-Rodríguez et al., 2019*) using a complementary approach. By performing a 'centromere re-activation' assay (*Tanaka et al., 2005*) the Tanaka lab showed that Bir1 deletion, and to a lesser extent Mcm21 depletion, reduced localization of Ipl1 at the centromere which was synergistic upon removal of both and the effect on Ipl1 localization correlated with the establishment of chromosome biorientation. This is consistent with our finding that the Ctf19 C-terminus has a role in accurate chromosome segregation and indicates that the Sli15-Ctf19 interaction contributes to the localization and stabilization of the CPC at the inner kinetochore (*Figure 7*).

Our findings also agree with the observations that the functionally active Aurora B pool is associated with the kinetochore rather than the centromere (*DeLuca et al., 2011*; *Bekier et al., 2015*; *Krenn and Musacchio, 2015*; *Hindriksen et al., 2017*). A recent study in humans demonstrated that a kinetochore-localized CPC pool lacking the INCENP CEN domain is sufficient to carry out error correction and biorientation, if cohesin removal, which was attributed to the loss of the CEN domain, is prevented (*Hengeveld et al., 2017*). Furthermore, retaining the human CPC at centromeres in anaphase resulted in the untimely recruitment of Bub1 and BubR1 (*Vázquez-Novelle and Petronczki, 2010*; *Vázquez-Novelle et al., 2014*) which suggests that centromere-localization of the CPC is required, and microtubule-association may not be sufficient, for fulfilling its function in the spindle assembly checkpoint and chromosome biorientation. The human CENP-OPQUR complex



**Figure 7.** Schematic model of the budding yeast kinetochore subunit architecture. The Okp1 core domain directly binds the essential motif of the Cse4 END suggesting that in contrast to humans, the dual recognition of Cse4-NCPs in *S. cerevisiae* is established by the essential inner kinetochore subunits Ame1/Okp1 and Mif2 through interaction with distinct Cse4 motifs. Together with the observation that Ctf19 associates with Sli15/Ipl1, further CPC interactions with the inner and outer kinetochore could be part of a kinetochore conformation that is dependent on Sli15[INCENP]. In line with the observed benomyl sensitivity of cells expressing Sli15ΔN as the only nuclear copy (*Figure 5C*), a recent study in Xenopus egg extracts found that CPC lacking the CEN domain of

*Figure 7 continued on next page*

*Figure 7 continued*

INCENP affected the correction of erroneous kinetochore-microtubule attachments (*Haase et al., 2017*). Centromere-targeting deficient CPC resulted in an imperfect inner kinetochore composition that failed to sense tension-loss and in intermediate Ndc80 phosphorylation levels that indicated the incapability of establishing a sharp phosphorylation gradient according to the spatial separation model. Flat Ndc80 phosphorylation levels could be sufficient for the non-selective turnover of erroneous kinetochore attachments, especially at budding yeast kinetochores which are attached to a single microtubule, unless cells are challenged by microtubule poisons.
DOI: https://doi.org/10.7554/eLife.42879.013

has recently been shown to promote accurate chromosome alignment by interaction with microtubules (*Pesenti et al., 2018*). If the observed interaction between the CPC and COMA is conserved in higher eukaryotes or is facilitated by other kinetochore proteins remains to be addressed.

In the spatial separation model the CPC is anchored at the centromere and substrate access of the Ipl1$^{Aurora\ B}$ kinase is regulated by tension-dependent intra-kinetochore stretching upon the biorientation of sister kinetochores. Whether the Ctf19-Sli15 interaction is required for CPC stabilization or for the precise positioning of Ipl1 activity at a distinct kinetochore conformation, competent for tension sensing and error correction, poses an interesting future question (*Figure 7*). Our findings place COMA at the center of kinetochore assembly in budding yeast and contribute to the molecular understanding of the fundamental process of how cells establish correct chromosome biorientation at the mitotic spindle.

# Materials and methods

## Key resources table

| Reagent type (species) or resource | Designation | Source or reference | Identifiers | Additional information |
|---|---|---|---|---|
| Gene (*S. cerevisiae*) | See *Supplementary file 5* | | | |
| Strain, strain background (*S. cerevisiae*) | S288c | | | |
| Strain, strain background (*E. coli*) | BL21(DE3) | New England Biolabs | C2527 | |
| Strain, strain background (*E. coli*) | DH10Bac | ThermoFisher | 10361012 | |
| Cell line (*S. frugiperda*) | SF21; *Spodoptera frugiperda* | ThermoFisher | 11497013 | |
| Cell line (*Trichoplusia ni*) | High five; *Trichoplusia ni* | ThermoFisher | B85502 | |
| Genetic reagent (*S. cerevisiae*) | See *Supplementary file 5* | | | |
| Antibody | Anti-FLAG M2 (mouse monoclonal) | Sigma-Aldrich | F1804 RRID:AB_262044 | 1:5000 |
| Antibody | Anti-PGK1 (mouse monoclonal) | Invitrogen | 22C5D8 RRID:AB_2532235 | 1:10000 |
| Antibody | goat anti-mouse IgG-HRP | Santa Cruz Biotechnology | sc-2005 RRID:AB_631736 | 1:10000 |
| Recombinant DNA reagent | See *Supplementary file 4* | | | |

*Continued on next page*

*Continued*

| Reagent type (species) or resource | Designation | Source or reference | Identifiers | Additional information |
|---|---|---|---|---|
| Peptide, recombinant protein | 3xFLAG peptide | Ontores | | |
| Peptide, recombinant protein | lambda phosphatase | New England Biolabs | P0753S | |
| Commercial assay or kit | Q5 Site-Directed Mutagenesis Kit | New England Biolabs | E0552S | |
| Chemical compound, drug | BS3-H12/D12 cross-linker | Creative Molecules | 001SS | |
| Chemical compound, drug | Iodoacetamide | Sigma-Aldrich | I6125 | |
| Chemical compound, drug | Lysyl Endopeptidase | FUJIFILM Wako Pure Chemical Corporation | 125–05061 | |
| Chemical compound, drug | Trypsin Sequencing Grade Modified | Promega | V5111 | |
| Chemical compound, drug | SYBR Gold | ThermoFisher | S11494 | |
| Chemical compound, drug | AMP-PNP | Santa Cruz Biotechnology | CAS 72957-42-7 | |
| Chemical compound, drug | Rapamycin | Invitrogen | PHZ1235 | |
| Chemical compound, drug | Concanavalin A from *Canavalia ensiformis* | Sigma-Aldrich | C2010 | |
| Chemical compound, drug | FuGENE HD Transfection Reagent | Sigma-Aldrich | E2311 | |
| Chemical compound, drug | cOmplete ULTRA EDTA-free Protease Inhibitor Cocktail | Roche | 5892953001 | |
| Chemical compound, drug | Ni-NTA Agarose | Qiagen | 30210 | |
| Chemical compound, drug | Strep-Tactin Superflow Plus Agarose | Qiagen | 30004 | |
| Chemical compound, drug | M2 anti-FLAG agarose | Sigma-Aldrich | A4596 | |
| Other | Sep-Pak tC18 cartridges | Waters | WAT054960 | |
| Other | PD-10 Desalting Columns | GE Healthcare | 17085101 | |
| Other | µ-Slide 8 Well | Ibidi | 80826 | |
| Software, algorithm | xQuest | (*Walzthoeni et al., 2012*) | | |
| Software, algorithm | xVis | (*Grimm et al., 2015*) | | |
| Software, algorithm | Fiji | (*Schindelin et al., 2012*) | | |
| Software, algorithm | Clustal Omega | (*Sievers et al., 2011*) | | |
| Software, algorithm | SoftWoRx | GE Healthcare | | |

## Chemical cross-linking and mass spectrometry of kinetochore complexes

The complex containing Cse4-NCP, Mif2, Ame1/Okp1, Ctf19/Mcm21, Chl4/Iml3 and MTW1c was assembled in solution. It was cross-linked using an equimolar mixture of isotopically light (hydrogen) and heavy (deuterium) labeled bis[sulfosuccinimidyl]suberate (BS3, H12/D12) (Creative Molecules) at a final concentration of 0.25–0.5 mM at 10℃ for 30 min. The reaction was quenched by adding ammonium bicarbonate to a final concentration of 100 mM for 10 min at 10 ℃. The sample was subjected to SEC on a Superose 6 Increase 10/300 GL column (GE Healthcare) and the fractions corresponding to the cross-linked complex were selected for the subsequent protein digest and mass spectrometry (see below).

The complex of Sli15-2xStrep-HA-6xHis/Ipl1 with Ame1/Okp1 and Ctf19/Mcm21 was assembled on Strep-Tactin Superflow agarose (Qiagen) by incubation at room temperature (RT), 1000 rpm for 1 hr in a thermomixer (Eppendorf). Unbound proteins were removed by washing three times with binding buffer [50 mM NaH$_2$PO$_4$(pH 8.0), 500 mM NaCl, 5% glycerol] and the complex was eluted in binding buffer containing 8 mM biotin. The eluted complex was re-isolated on Ni-NTA beads (Qiagen), washed twice with binding buffer and then cross-linked by resuspending the protein bound beads in BS3 cross-linker at a final concentration of 0.25–0.5 mM at 30℃ for 30 min. The cross-linking reaction was stopped by adding ammonium bicarbonate to a final concentration of 100 mM for 20 min at 30℃.

Cross-linked samples were denatured by adding two sample volumes of 8 M urea, reduced with 5 mM TCEP (ThermoFisher) and alkylated by the addition of 10 mM iodoacetamide (Sigma-Aldrich) for 40 min at RT in the dark. Proteins were digested with Lys-C (1:50 (w/w), FUJIFILM Wako Pure Chemical Corporation) at 35℃ for 2 hr, diluted with 50 mM ammonium bicarbonate, and digested with trypsin (1:50 w/w, Promega) overnight. Peptides were acidified with trifluoroacetic acid (TFA) at a final concentration of 1% and purified by reversed phase chromatography using C18 cartridges (Sep-Pak, Waters). Cross-linked peptides were enriched on a Superdex Peptide PC 3.2/30 column using water/acetonitrile/TFA (75/25/0.1, v/v/v) as mobile phase at a flow rate of 50 μl/min and were analyzed by liquid chromatography coupled to tandem mass spectrometry (LC-MS/MS) using an Orbitrap Elite instrument (ThermoFisher). Fragment ion spectra were searched and cross-links were identified by the dedicated software *xQuest* (*Walzthoeni et al., 2012*). The results were filtered according to the following parameters: Δscore $\leq$ 0.85, MS1 tolerance window of −4 to 4 ppm and score $\geq$ 22. The quality of all cross-link spectra passing the filter was manually validated and cross-links were visualized as network plots using the webserver *xVis* (*Grimm et al., 2015*).

## Electrophoretic mobility shift assay

Reconstituted nucleosomes (0.5 μM) were mixed in a 1:2 molar ratio with the respective protein complexes in a buffer containing 20 mM Hepes (pH 7.5) and incubated for 1 hr on ice. The interaction was analyzed by electrophoresis at 130 V for 70–90 min on a 6% native polyacrylamide gel in a buffer containing 25 mM Tris and 25 mM boric acid. After electrophoresis, gels were stained with SYBR Gold (ThermoFisher).

## Analytical size exclusion chromatography for interaction studies

Analytical SEC experiments were performed on a Superdex 200 Increase 3.2/300 or a Superose 6 Increase 3.2/300 column (GE Healthcare). To detect the formation of a complex, proteins were mixed at equimolar ratios and incubated for 1 hr on ice before SEC. All samples were eluted under isocratic conditions at 4℃ in SEC buffer [50 mM HEPES (pH 7.5), 150 mM NaCl, 5% glycerol]. Elution of proteins was monitored by absorbance at 280 nm. 100 μl fractions were collected and analyzed by SDS-PAGE and Coomassie staining.

## In vitro protein binding assay of Sli15/Ipl1 to Ame1/Okp1 and/or Ctf19/Mcm21

Phosphorylated or non-phosphorylated wild-type or mutant Sli15-2xStrep-HA-6xHis/Ipl1 was immobilized on Strep-Tactin Superflow agarose (Qiagen). For prephosphorylation, Sli15/Ipl1 was incubated at 30℃ for 30 min in the presence of 3 mM MgCl$_2$ and 3 mM ATP. Samples for non-phosphorylated Sli15/Ipl1 were treated the same way, but instead of 3 mM ATP the non-

hydrolysable analog AMP-PNP (Santa Cruz Biotechnology) was applied. To remove basal phosphorylation, Sli15/Ipl1 was treated with lambda phosphatase (New England Biolabs) at 30°C for 30 min. Subsequently, non-phosphorylated as well as phosphorylated or dephosphorylated Sli15/Ipl1 complexes were washed three times with binding buffer [50 mM $NaH_2PO_4$(pH 8), 120 mM NaCl, 5% glycerol].

Testing of binding between Ame1/Okp1, Ctf19/Mcm21 and Sli15/Ipl1 was performed in binding buffer at 4°C, 1000 rpm for 1 hr in a thermomixer (Eppendorf). Unbound proteins were removed by washing three times with binding buffer. The complexes were either eluted with 8 mM biotin in 50 mM $NaH_2PO_4$(pH 8), 500 mM NaCl, 5% glycerol or by boiling in 2x SDS loading buffer.

To quantify the ratios of bound proteins to the bait protein SDS page band intensities were analyzed by using the Fiji software (*Schindelin et al., 2012*).

## Amino acid sequence alignment

Multiple sequence alignment of Cse4 or Okp1 protein sequences from interrelated budding yeast species was conducted with Clustal Omega (*Sievers et al., 2011*). Only protein sequences with the highest similarity to *S. cerevisiae* Cse4 or *S. cerevisiae* Okp1 as determined by a protein BLAST search were included in the search. In addition three mammalian and the *Schizosaccharomyces pombe* homologous CENP-A protein sequences were included in the Cse4 alignment.

## Yeast strains and methods

All plasmids and yeast strains used in this study are listed in *Supplementary file 4* and *Supplementary file 5*, respectively. Yeast strains were created in the S288c background. The generation of yeast strains and yeast methods were performed by standard procedures. The anchor-away technique was performed as previously described (*Haruki et al., 2008*).

For anchor-away rescue experiments, the respective promoters and coding sequences were PCR amplified from yeast genomic DNA and cloned into the vector pRS313 either via the Gibson assembly or the restriction/ligation method. In order to artificially target Sli15ΔN2-228 to the kinetochore, the individual promoters and genes were PCR amplified and the respective gene fusions [*CTF19, AME1, OKP1, CTF3, CNN1, MIF2, DSN1, MTW1*]-[*SLI15ΔN2-228*]-[*6xHis-7xFLAG*] (*Supplementary file 4*) were generated and cloned into pRS313 using the Gibson assembly reaction The same strategy was applied in order to generate the *CTF19* or *CTF19ΔC* gene fusions to *AME1* or *OKP1*, respectively (*Supplementary file 4*).

The individual deletion mutants were generated using the Q5 site-directed mutagenesis kit (New England Biolabs). The rescue constructs were transformed into Cse4-, Ctf19-, Okp1-, or Sli15 anchor-away strains (*Supplementary file 5*) and cell growth was tested in 1:10 serial dilutions on YPD plates in the absence or presence of rapamycin (1 μg/ml) at 30°C for 3 days.

## Minichromosome loss assay

The Ctf19 anchor-away strain containing a minichromosome (*pYCF1/CEN3.L*) (*Spencer et al., 1990*) and the Ctf19 anchor-away strains containing a minichromosome (*pYCF1/CEN3.L*) and the respective rescue plasmid were grown overnight in selective medium (-Ura selecting for the minichromosome, or –His/-Ura selecting for the rescue plasmid and the minichromosome) and then diluted into YPD medium and cultured for 4 hr. The yeast cultures were then plated onto synthetic medium containing rapamycin (1 μg/ml) and low (6 μg/ml) adenine to enhance the red pigmentation (*Hieter et al., 1985*) and incubated for 3 days at 30°C. Colonies retaining the minichromosome are white, and loss events result in the formation of red/red sectored colonies. The minichromosome loss frequency was quantified by determining the percentage of red/red sectored colonies in relation to the total colony number (white and red/red sectored) of three biological replicates.

## Western blot analysis

For western blot analysis an equivalent of 10 $OD_{600}$ of cells logarithmically grown in liquid culture was collected by centrifugation at 3140 x g for 5 min at RT and the pellet was washed once with aqua dest. For protein extraction, the pellet was resuspended in 1 ml ice-cold 10% trichloroacetic acid and incubated on ice for 1 hr. Samples were pelleted at 20000x g for 10 min, 4°C and washed twice with ice-cold 95% ethanol. Pellets were air-dried and resuspended in 100 μl 1x SDS-PAGE

sample buffer containing 75 mM Tris (pH 8.8). Samples were boiled (10 min, 95°C) and centrifuged at 10800 x g for 3 min at RT and supernatants were separated on 10% or 15% (Cse4 containing samples) SDS-PAGE gels. Immunoblotting was performed with the following antibodies: Anti-FLAG M2 (Sigma-Aldrich), Anti-PGK1 (ThermoFisher) and visualized by HRP-conjugated anti-mouse secondary antibodies (Santa Cruz).

## Live cell microscopy

For localisation analysis of endogenously tagged Ctf19-GFP and Ctf19ΔC-GFP proteins, cells were grown in synthetic medium without tryptophan at 30°C. For localisation analysis of ectopically expressed Ctf19-Okp1-GFP and Ctf19ΔC-Okp1-GFP proteins in the Ctf19-anchor-away (Ctf19-FRB) strain, cells were grown in selective medium (–His/-Trp) until $OD_{600}$ ~0.4, then rapamycin (1 µg/ml) was added and cells were grown for another 3 hr at 30°C. For imaging cells were immobilized on concanavalin-A (Sigma-Aldrich) coated slides (Ibidi). Microscopy was performed using a DeltaVision microscopy system (Applied precision) with a Olympus IX71 microscope controlled by softWoRx software (GE Healthcare). Images were processed using Fiji (*Schindelin et al., 2012*).

## Protein expression and purification

Expression constructs for 6xHis-Chl4/Iml3, 6xHis-Cnn1/Wip1-1xFlag, 6xHis-Nkp1/Nkp2 and Mhf1/Mhf2-1xStrep were created by amplification of genomic DNA and cloned into pETDuet-1 vector (Novagen). Expression was performed in BL21 (DE3) cells (New England Biolabs). Cells were grown at 37°C until $OD_{600}$ 0.6, followed by induction with 0.5 mM IPTG for Chl4/Iml3 or 0.2 mM IPTG for all other protein expressions. Protein expression was induced overnight at 18°C, or for 3 hr at 23°C, respectively.

Cells were lysed using a French Press in lysis buffer [50 mM Hepes (pH 7.5), 400 mM NaCl, 3% glycerol, 0.01% Tween20 and cOmplete ULTRA EDTA-free Protease Inhibitor Cocktail (Roche)]. 6xHis-tagged proteins were purified using Ni-NTA agarose (Qiagen), whereby 30 mM imidazole were added to the lysis buffer in the washing step, followed by protein elution in 50 mM Hepes pH 7.5, 150 mM NaCl, 300 mM imidazole, and 5% glycerol. Strep-tag purification was performed using Strep-Tactin Superflow agarose (Qiagen) and eluted in a buffer containing 50 mM Hepes (pH 7.5), 150 mM NaCl, 8 mM biotin and 5% glycerol.

Buffer exchange into a buffer containing 50 mM Hepes (pH 7.5), 150 mM NaCl and 5% glycerol was performed using a Superdex 200 HiLoad 16/60 column (GE Healthcare) for Chl4/Iml3 and Cnn1/Wip1 or using a PD10 desalting column (GE Healthcare) for Nkp1/2 and Mhf1/2 protein complexes.

## Ame1/Okp1 expression and purification

Ame1-6xHis/Okp1 wild-type and mutant protein expression and purification in *E. coli* was performed as described previously (*Hornung et al., 2014*).

## In vitro reconstitution of Cse4- and H3-NCPs

Octameric Cse4 and H3 containing nucleosomes were *in vitro* reconstituted from budding yeast histones which were recombinantly expressed in *E. coli* BL21 (DE3) and assembled on 167 bp of the 'Widom 601' nucleosome positioning sequence according to a modified protocol (*Turco et al., 2015*).

## Affinity-purification of recombinant protein complexes from insect cells

C-terminal 6xHis-6xFLAG-tags on Mcm21, Mif2, Dsn1, Mcm16 and C-terminal 2xStrep- tags on Sli15 were used to affinity-purify Ctf19/Mcm21, Mif2, MTW1c, CTF3c and Sli15/Ipl1 complexes. Open reading frames encoding the respective subunits were amplified from yeast genomic DNA and cloned into the pBIG1/2 vectors according to the biGBac system (*Weissmann et al., 2016*). The pBIG1/2 constructs were used to generate recombinant baculoviral genomes by Tn7 transposition into the DH10Bac *E. coli* strain (ThermoFisher) (*Vijayachandran et al., 2011*). Viruses were generated by transfection of Sf21 insect cells (ThermoFisher) with the recombinant baculoviral genome using FuGENE HD transfection reagent (Promega). Viruses were amplified by adding transfection supernatant to Sf21 suspension cultures. Protein complexes were expressed in High Fiveinsect cell (ThermoFisher) suspension cultures.

For purification of FLAG-tagged kinetochore complexes, insect cells were extracted in lysis buffer [50 mM Tris (pH 7.5), 150 mM NaCl, 5% glycerol] supplemented with cOmplete ULTRA EDTA-free Protease Inhibitor Cocktail (Roche) using a Dounce homogenizer. Cleared extracts were incubated with M2 anti-FLAG agarose (Sigma-Aldrich) for 2 hr, washed three times with lysis buffer and eluted in lysis buffer containing 1 mg/ml 3xFLAG peptide (Ontores).

High Five cells expressing Strep-tagged Sli15/Ipl1 were lysed in 50 mM $NaH_2PO_4$(pH 8.0), 300 mM NaCl, 5% glycerol supplemented with cOmplete ULTRA EDTA-free Protease Inhibitor Cocktail (Roche). Subsequent to incubating the cleared lysates with Strep-Tactin Superflow agarose (Qiagen), protein bound beads were washed three times with lysis buffer and the bound protein complex was eluted in lysis buffer containing 8 mM biotin. FLAG peptide or biotin was either removed via PD10 desalting columns (GE Healthcare) or SEC using a Superdex 200 HiLoad 16/60 column (GE Health-care) and isocratic elution in lysis buffer.

## Acknowledgements

We are grateful to Andrea Musacchio (MPI Dortmund) and Stefan Westermann (University of Essen) for discussions and sharing reagents. We thank Wolfgang Zachariae (MPI Munich) for help with fluorescence microscopy. JFH and GH were funded by the Graduate School (GRK 1721) and MP and VS were funded by the Graduate School (Quantitative Biosciences Munich) of the German Research Foundation (DFG). LDG was a recipient of a DOC Fellowship of the Austrian Academy of Sciences and AK was funded by ERC Grant 281354 (NPC GENEXPRESS). FH was supported by the European Research Council (ERC-StG no. 638218), the Human Frontier Science Program (RGP0008/2015), by the Bavarian Research Center of Molecular Biosystems and by an LMU excellent junior grant.

## Additional information

### Funding

| Funder | Grant reference number | Author |
| --- | --- | --- |
| Deutsche Forschungsgemeinschaft | Graduate School Quantitative Biosciences Munich | Mia Potocnjak Victor Solis-Mezarino |
| Deutsche Forschungsgemeinschaft | Graduate School GRK 1721 | Götz Hagemann Franz Herzog |
| Austrian Academy of Sciences | DOC Fellowship | Laura D Gallego |
| European Research Council | 281354 (NPC GENEXPRESS) | Alwin Köhler |
| European Research Council | ERC-StG MolStruKT, no. 638218 | Franz Herzog |
| Human Frontier Science Program | RGP0008/2015 | Franz Herzog |
| Bavarian Research Center for Molecular Biosystems | | Franz Herzog |
| Ludwig-Maximilians-Universität München | Excellent Junior grant | Franz Herzog |

The funders had no role in study design, data collection and interpretation, or the decision to submit the work for publication.

### Author contributions

Josef Fischböck-Halwachs, Sylvia Singh, Formal analysis, Investigation, Methodology, Writing—original draft; Mia Potocnjak, Formal analysis, Methodology; Götz Hagemann, Victor Solis-Mezarino, Formal analysis, Investigation; Stephan Woike, Formal analysis, Investigation, Methodology; Medini Ghodgaonkar-Steger, Jessica Andreani, Formal analysis; Florian Weissmann, Laura D Gallego, Julie Rojas, Alwin Köhler, Methodology; Franz Herzog, Conceptualization, Funding acquisition, Writing—original draft

## Author ORCIDs
Jessica Andreani (iD) https://orcid.org/0000-0003-4435-9093
Franz Herzog (iD) https://orcid.org/0000-0001-8270-1449

## Decision letter and Author response
Decision letter https://doi.org/10.7554/eLife.42879.025
Author response https://doi.org/10.7554/eLife.42879.026

# Additional files

### Supplementary files
• Supplementary file 1. Inter- and intra-protein cross-links detected on in vitro reconstituted Cse4 containing nucleosomes interacting with the kinetochore complexes Ame1/Okp1, Ctf19/Mcm21, Mif2, Chl4/Iml3 and MTW1c.
DOI: https://doi.org/10.7554/eLife.42879.014

• Supplementary file 2. Inter- and intra-protein cross-links detected on in vitro reconstituted Sli15/Ipl1 interacting with the inner kinetochore proteins Ctf19, Okp1, Ame1 and Mcm21 (COMA).
DOI: https://doi.org/10.7554/eLife.42879.015

• Supplementary file 3. Predicted and experimentally annotated protein domains and motifs depicted in protein cross-link networks.
DOI: https://doi.org/10.7554/eLife.42879.016

• Supplementary file 4. Plasmids used in this study.
DOI: https://doi.org/10.7554/eLife.42879.017

• Supplementary file 5. Yeast strains used in this study.
DOI: https://doi.org/10.7554/eLife.42879.018

• Transparent reporting form
DOI: https://doi.org/10.7554/eLife.42879.019

### Data availability
The mass spectrometry raw data was uploaded to the PRIDE Archive and is publicly available through the following identifiers: PXD011235 (COMA-Sli15/Ipl1); PXD011236 (CCAN).

The following datasets were generated:

| Author(s) | Year | Dataset title | Dataset URL | Database and Identifier |
|---|---|---|---|---|
| Josef Fischböck-Halwachs, Sylvia Singh, Mia Potocnjak, Götz Hagemann, Victor Solis-Mezarino, Stephan Woike, Medini Ghodgaonkar-Steger, Florian Weissmann, Laura D. Gallego, Julie Rojas, Jessica Andreani, Alwin Köhler, Franz Herzog | 2019 | COMA-CPC | https://www.ebi.ac.uk/pride/archive/projects/PXD011235 | PRIDE, PXD011235 |
| Josef Fischböck-Halwachs, Sylvia Singh, Mia Potocnjak, Götz Hagemann, Victor Solis-Mezarino, Stephan Woike, Medini Ghodgaonkar-Steger, Florian Weissmann, Laura D. | 2019 | CCAN | https://www.ebi.ac.uk/pride/archive/projects/PXD011236 | PRIDE, PXD011236 |

Gallego, Julie Rojas,
Jessica Andreani,
Alwin Köhler, Franz
Herzog

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
