## [Decision Letter]

Thank you for submitting your article "The COMA complex is required for positioning Ipl1 activity proximal to Cse4 nucleosomes in budding yeast" for consideration by *eLife*. Your article has been reviewed by three peer reviewers, including Jennifer G. DeLuca as the Reviewing Editor and Reviewer #3, and the evaluation has been overseen by Anna Akhmanova as the Senior Editor. The following individual involved in review of your submission has agreed to reveal their identity: Sue Biggins (Reviewer #1).

The reviewers have discussed the reviews with one another and the Reviewing Editor has drafted this decision to help you prepare a revised submission.

Summary:

In this manuscript, the authors investigate the organization of the budding yeast kinetochore using a combination of biochemical reconstitution and chemical crosslinking/mass spec (XMLS). In doing so, they identify novel interactions within the kinetochore and additionally, they map a direct binding site for the Chromosomal Passenger Complex (CPC) to the inner kinetochore. Specifically, they make two key findings: (1) They demonstrate an interaction between the N-terminus of Cse4 (the yeast homolog of CENP-A) and Okp1 (a member of the COMA complex). The interaction interfaces they identify are essential, underscoring the role these interactions play in kinetochore assembly. (2) They find that the C-terminal RWD of Ctf19 (also a member of the COMA complex) binds the CPC component Sli15 (the yeast homolog of INCENP). This is a major finding because it had been previously shown that CPC localization to the inner centromere is not essential for viability. Based on genetic experiments, the authors demonstrate that this may be due to a second CPC binding site in Ctf19 that can support viability when the inner centromere localization is disrupted. In addition, the authors show that the interaction between Ctf19 and Sli15 is regulated by inhibitory Ipl1/Aurora B phosphorylation. Overall, the reviewers were positive about the work and agree that this study provides important new information regarding the organization of eukaryotic kinetochores and defines a new mechanism for CPC recruitment to the kinetochore region. The reviewers also raise concerns regarding interpretation of the major findings, and in particular, three areas require significant attention.

First, the authors place great weight on the apparent absence of direct contact between Chl4 and Cse4. From this, they wish to infer large structural differences between the human and yeast inner kinetochore. These claims depend on the reconstitution recapitulating faithfully the assembly as it occurs inside the yeast nucleus. This may not be the case, for several reasons: (a) use of 601 DNA instead of a true centromere, (b) absence of post-translational regulation, and (c) lack of ordered assembly afforded by a single-mixture approach. Indeed, if the authors have isolated a true inner kinetochore assembly upon Cse4 (which would be a substantial step forward), they should show that it survives purification with the appropriate stoichiometry. Without such evidence, the discussion of XLMS findings must be tempered appropriately. It is also noted that in Figure 2 of Weir et al., in which similar XLMS studies were done with human proteins, there is only a single crosslink between CENP-N and CENP-A, and that link comes from the "wrong end" (the C-terminal end) of CENP-A. We nonetheless know from the work of Musacchio and Luger that CENP-N binds selectively with CENP-A nucleosomes. So, the absence of a crosslink cannot be taken as evidence that the structures differ in any major way. The conclusion that there is some important structural difference between human and yeast should be removed in favor of a far more nuanced statement. There are evidently some differences, either in assembly hierarchy or in pairwise affinities; in view of the divergence of sequences between orthologs, the absence of Nkp1/2 in metazoans, etc., it would be very surprising if there were not. But the current data do not tell us anything about those issues, subtle or otherwise.

Second, the authors point to the inability of kinetochore components besides Okp1 and Ame1 to rescue the viability of Sli15dN cells upon depletion of Ctf19 as evidence that Sli15 must be positioned at the inner kinetochore to carry out its function. A true test of this hypothesis would require a Ctf19 allele that disrupts Sli15 binding but does not affect outer kinetochore assembly. Ctf19 depletion causes removal of much of the outer kinetochore, making the presented experiment a poor test of the model. Furthermore, the authors do not present evidence that disruption of this specific interaction leads to the expected consequences in cells, namely defects in chromosome bi-orientation or error correction. This would go a long way in strengthening the authors' conclusions.

Third, numerous studies have provided evidence for multiple distinct populations of the CPC in mitotic cells, including at the inner centromere, the kinetochore, spindle microtubules, and the spindle midzone. The authors, in identifying a binding site for CPC in the inner kinetochore, seem to have made major progress towards separating these CPC populations. Unfortunately, discussion of this issue is missing. That Sli15dN does not support full viability on benomyl (Figure 5E) seriously confounds the issue. How do the authors reconcile this finding with that of Campbell and Desai, which shows the opposite? What does this imply about the existence of distinct CPC populations? Which one might be responsible for tension sensing? Related to this point, there is confusion throughout the manuscript as to whether the authors are intending to differentiate between specific populations of the CPC, and whether they attribute loss of function/viability to one population or another. This becomes a critical issue when interpreting how different pools of the CPC are recruited to and functionally utilized at the centromere and kinetochore, a point that seems central to this study.

These are complex issues, and full answers may not be available. Nevertheless, the findings presented provide new information that will allow many laboratories to ask specific questions about CPC function in chromosome condensation, tension sensing, and kinetochore assembly in mitosis and meiosis. For this reason, the reviewers are in support of publication, and they agree that these issues can be addressed through a limited number of experimental revisions as well as major revisions to the text and figures.

Essential revisions (experimental):

1) The discovery of a potentially new binding site for the CPC at the inner centromere is an important advance, but there is no demonstration that this interaction site exists in vivo, nor what the phenotypes of impairing this interaction are. While the tethering experiments are compelling, the authors "anchor-away" the entire Ctf19 protein, which leaves open the possibility that the phenotypes are due to multiple Ctf19 functions. An attempt to analyze Ipl1 localization in vivo or show that error correction requires the interaction between Ipl1 and Ctf19 would strengthen the findings. Ideally, the authors would use their conditional system to show that the mutation of the Ctf19 binding site eliminates all Ipl1 from kinetochores when the Sli15 N-terminus is mutated. At a minimum, the authors should analyze bi-orientation in a Sli15 mutant in the Ctf19 binding site to show that the cells are dying due to a bi-orientation defect as expected.

2) It is difficult to see some of the proposed interactions in Figure 4B -- quantification of this binding experiment would be useful. An additional issue with this figure, which is crucial to the manuscript as a whole, is that pulled-down material for single inputs alone (Sli15/Ipl1, Ame1/Okp1, Ctf19/Mcm21, etc.) are not shown, making the experiment difficult to interpret. These controls should be included.

3) The authors demonstrate that phosphorylation of Sli15 by Ipl1 negatively regulates CPC binding to Ctf19, but there is no indication that this is physiologically relevant. If possible, the authors should test if the phospho-mutants (or mimics) have an effect on viability or chromosome segregation.

4) Related to the above point, the authors expressed Sli15-Ipl1 in insect cells and show basal phosphorylation. Might there be some level of phosphorylation that impacts COMA interaction before ATP treatment? It would be informative to test if the interaction is positively affected after treatment with lambda phosphatase.

5) Figures 2D, 5D and 5E: Immunoblots to demonstrate protein expression should be included.

Essential revisions (non-experimental):

1) "Epigenetically marked" is inappropriate for budding yeast.

2) Introduction, third paragraph: The end of this paragraph needs to be rewritten. The authors say that Campbell showed Sli15dN could still localize to kinetochores. They then discuss centromere-targeting deficient Sli15/Ipl1. Which is it? If the authors are referring to distinct Sli15/Ipl1 pools (kinetochore versus centromere?), then they should make the reference to distinct pools explicit and substantiate them with the proper references.

3) Introduction, last paragraph: Topology is the wrong word.

4) "The interaction is not mediated by AT-rich DNA sequences" should say, "The interaction does not require AT-rich DNA sequences."

5) Subsection “The Ame1/Okp1 heterodimer selectively binds Cse4 containing nucleosomes”, last paragraph: The authors describe crosslinks between Mif2n and MIND. Did they use the Dsn1-2D mutant? If not, they should comment on the prevalence of these crosslinks relative to expectations. Why do they see them at all?

6) Results, subsection “The Ame1/Okp1 heterodimer selectively binds Cse4 containing nucleosomes” last paragraph: "no direct interaction" is not formally correct. See Schmitzberger, 2017 supplement.

7) The following is not a correct conclusion: "Chl4, in contrast to its human ortholog, does not recognize Cse4-NCPs." The strongest claim that can be made is that the authors have not found evidence supporting this idea. Nor do the published XLMS data for the human ortholog provide such evidence -- only direct binding and a structure. One could imagine a number of reasons for absence of crosslinks: no lysine pairs accessible to BS3 link Chl4 and Cse4; 601 is an inappropriate DNA substrate to use for these studies; post-translational modifications are required for Chl4-Cse4 engagement. In short, although the data are clear on many points and extremely valuable overall, they cannot prove the negative.

8) "END domain" is redundant.

9) "suggests that recruitment of the Ame1/Okp1 heterodimer" This should be rewritten to clarify the point that the suggested recruitment relationship between Cse4-END and Okp1-Ame1 is the speculation here. This modification is important; although the evidence presented is consistent with recruitment (Okp1-Ame1 by Cse4), no conclusive test of such a relationship is presented. An alternative hypothesis – that Cse4-END regulates an essential function of either Okp1-Ame1 or Cse4 – is equally likely given the data.

10) "Ctf19 is the primary Sli15/Ipl1 interaction site […]" This statement is too strong. Instead, it should read "is required for Sli15/Ipl1 interaction"

11) Subsection “Tethering Sli15ΔN selectively to COMA rescues the synthetic lethality of a sli15ΔN mutant upon Ctf19 depletion”, last paragraph: This paragraph should be split into at least two and possibly three or four paragraphs.

12) Subsection “Tethering Sli15ΔN selectively to COMA rescues the synthetic lethality of a sli15ΔN mutant upon Ctf19 depletion”, last paragraph: That Ctf19dN is not synthetic with Sli15dN is both interesting and important. If impaired cohesin loading does not explain the synthetic interaction with Ctf19d (as apparently it does not), how would the authors interpret the finding from Campbell and Desai that Ctf19d is synthetic with Sli15dN? When this finding was originally presented, it was framed as evidence for the cohesin hypothesis, which here seems discredited. A single sentence in the Discussion section would be appropriate.

13) "Endogenous Sli15dN could not rescue in trans […]" and "In contrast, deletion of the SAH domain in Ame1- and Okp1-Sli14dNdSAH fusions was not lethal and could be rescued in trans." These statements are vague. Could not be rescued by what in trans? I suspect this is a reference to later Campbell work (Fink et al., 2017) showing that induced dimerization of Sli15 rescues viability of cells expressing more perturbative Sli15dN alleles (d2-500, for instance). If so, this should be made explicit and the reference included here. Even so, I don't quite see how the current findings address the later Campbell observations in any substantial way.

14) "[…] suggesting that the SAH domain is not required for the function of the CPC proximal to the centromere." What do the authors mean by "proximal to the centromere?" My understanding is that CPC has multiple functions, likely corresponding to distinct modes of localization: (1) KT-bound CPC localizes, as shown here, through COMA; 2) MT/spindle-bound CPC localizes through Sli15 and is regulated by CDK; 3) CPC targets chromatin through a poorly-defined localization pathway that probably differs in yeast and vertebrates and that corresponds to H3 phosphorylation throughout the pericentromere. This third pathway appears to be disabled by the Sli15dN allele, but this property has been a source of some confusion, because whether CPC localizes to the centromere seems to depend on MT attachment status and the cell cycle. The third pool of CPC must sense cell-cycle regulation as it appears to be involved in chromosome condensation. It is unclear which CPC population the authors refer to in the statement quoted above. If it's the third, then they need to look at pH3. If they would rather not address this difficult topic, then the sentence should be rewritten to clarify this point.

15) Subsection “Tethering Sli15ΔN selectively to COMA rescues the synthetic lethality of a sli15ΔN mutant upon Ctf19 depletion”, last paragraph: The text references Figure 5E but makes no mention of the benomyl experiment. How do the authors interpret this experiment (see also comments in the summary)?

16) Subsection “The Ame1/Okp1 heterodimer directly links Cse4 nucleosomes to the outer kinetochore”, first paragraph: condense this paragraph. Don't rehearse all the data: just summarize the results in 2-3 sentences and end with the conclusion stated at the end of the aforementioned paragraph.

17) "As MTW1c recruitment by Mif2 and Cnn1 are redundant […]" There are at least two issues with this statement. First, there is no definitive evidence that yeast Cnn1 recruits MIND. There is, however, good evidence that Cnn1 interacts with Spc24/25. Second, although the two Ndc80 recruitment modules (Cnn1N and MIND via Mif2N) appear redundant, they likely serve distinct functions at different times during the cell cycle. Indeed, the two recruitment modules appear to be tightly regulated by phosphorylation, and different sets of kinases regulate the relevant components.

18) "Ame1/Okp1 is the sole essential link of the centromeric nucleosomes to the outer kinetochore […]" This statement is too strong. First, the authors have not conclusively shown that the phenotypes they observe are due to impaired Okp1-Ame1 recruitment. Second, the double mutant Cnn1d Mif2dN strain is sick and especially so at elevated temperature (Hornung, 2014). Would this not also be an essential link? Third, Mif2N and Ame1N appear to be two parts of the same connection to MIND. The authors must be explicit about necessary and sufficient conditions here, otherwise the above statement is vague and potentially misleading.

19) "is consistent with our finding of Chl4/Iml3 being positioned distal from Cse4" The authors have not positioned Chl4/Iml3 with respect to Cse4. Instead, they have not found any conclusive evidence that Chl4/Iml3 and Cse4 are near each other. Surely, Chl4 binds DNA. Would this DNA binding be carried out distal to the Cse4 nucleosome? How distal?

20) "The distinct architectures of vertebrate and budding yeast inner kinetochores […]" The authors have not shown conclusively that the "architectures" differ substantially. Although they have shown that Cse4/CENP-A engagement may differ, it is unclear by how much they differ, to what extent experimental differences contributed to the perceived divergence, and whether the apparent differences actually reflect differential regulation in yeast and vertebrate cells. The above statement should be amended to reflect this uncertainty. If the authors wish to state that there are substantial differences, they should present this as a hypothesis and not a direct observation arising from their data. In general, this section would benefit from a direct comparison with the crosslinking data presented in the Weir paper (human kinetochore crosslinking mass-spectrometry), of which Herzog was a co-author. An overall assessment of the ways these maps agree and differ would be helpful. In any case, the last three sentences of the subsection “Dual recognition of Cse4 at point centromeres by a CCAN architecture distinct from vertebrate regional centromeres” are unduly speculative, so if the authors do not wish to make the Weir comparison explicitly, they should delete most or all of those sentences, as they distract from the genuinely novel findings about CPC tethering.

21) "Ctf19 or Mcm21 become essential for viability once Sli15 loses its ability to be recruited to the inner centromere" The authors struggle to differentiate between CPC localization modes. In this case, the error is grave, as Campbell and Desai showed that the Sli15dN mutant does indeed localize to centromeres, although less efficiently than Sli15WT. Further, it seems the point of this manuscript is that Ctf19 supports centromeric Sli15 localization in the absence of Sli15N. Conceivably, the authors mean to differentiate between KT and inner centromere localization, but my understanding is that the "inner centromere" (what I would call pericentromeric) localization of CPC is poorly worked out and has not been knowingly experimentally separated from KT-mediated localization.

22) Subsection “COMA and its role in positioning Sli15/Ipl1 at the inner kinetochore”, second paragraph: "in trans" This is vague. Just removing these two words would clarify substantially and adding "by endogenous Sli15dN through its SAH domain".

23) "[…] that facilitates the selective recognition and destabilization of erroneous microtubule attachments upon loss of tension." This statement is not only unsupported by the data presented, it is at odds with Figure 5F. If Ctf19-bound CPC is the pool that does tension sensing, then the Sli15dN allele should not cause cells to be sicker than WT on benomyl. Instead, Sli15dN, when present as the only nuclear copy of Sli15, does not support viability on benomyl. This suggests the authors have found a CPC function at the inner KT that is distinct from correcting MT-KT attachment errors and that Sli15N is responsible for tension sensing. Does this KT-specific function reflect a role for CPC in kinetochore assembly? This should be discussed in the text. Might this reflect the possibly that distinct CPC populations do Dsn1 phosphorylation and DASH phosphorylation? Is there another CPC substrate at the inner-KT that is important and that we don't yet know about? Finally, with regard to this point, how do the authors reconcile this finding with Supplementary Figure 2C from Campbell and Desai? Those authors did not see a viability defect when they looked at sli15dN on benomyl. Could this reflect differences in the experimental setup?

24) "required for cohesin loading" A minor error: the Ctf19 fragment is required to recruit Scc2/4 to centromeres in late G1 but not for general cohesin loading.

25) Figure 1A: The concentration of NCP (and consequently the tested protein) used is not listed anywhere in the manuscript. This also applies to Figure 3B. Usually, EMSAs are done with very low concentrations of the labeled reagent (DNA here) and a range of concentrations for the tested binder. The authors have therefore set these up slightly unconventionally. That's fine, as long as they specify the conditions clearly.

26) Figure 1B: Why are Nkp1/2 apparently so large? Why are Cnn1/Wip1 so small, and where is Cnn1? What is the double band at ~25 kDa for both nucleosome preparations?

27) Figure 2A: What do the symbols beneath the alignment mean? They are not referenced in the figure legend. The protein shown in the alignment (also for B) needs to be specified in the figure so that a fast reader does not have to consult the legend.

28) Figure 2C: What are the "tags" referred to in the figure? These appear to be different preparations than those used in Figure 1, but there is only one type of co-expression vector listed in the supplement.

If the authors remove panel F, as requested below, then panel 2C could be rearranged to make it more readable. Specifically, the Cse4 mutant being tested should be written in larger font, and the cartoons showing each test should be removed.

29) Figure 2F: Remove this panel. It confuses and does not add meaningful information. Why are Cse4END and Okp1 both red? Why are structural models shown for both peptides, as no published experimentally-determined structure exists for either?

30) Figure 3A: The protein being shown in the figure needs to be specified clearly in the figure itself. Can the colors for the three lines be either all the same or more different? It's difficult to read as is. Also, I think including the ∆ is unnecessary here, although it is important in panel B.

31) Figure 3B: The ladder should be removed.

32) Figure 4B: Why does the full COMA pull down less efficiently than Ctf19-Mcm21 alone? Why did the authors not test Sli15-Ipl1 phosphorylation with Ctf19-Mcm21 alone?

I would also suggest putting the legend beneath the gels to make the results easier to interpret.

33) Figure 4D: What is the lower Ipl1 band/fragment, and why is it apparently missing from the pulled-down material for P4, P6, and P7?

34) Figure 6: Do the authors really believe Mhf1/2 participate in a nucleosome-like particle with Cnn1-Wip1? This should be discussed.

35) How does this work affect one's reading of Boekmann et al., which describes phosphorylation sites in Cse4N (thought to be Ipl1 sites) and a synthetic interaction between ipl1-ts and mutations at these phosphorylated residues?

36) The authors should cite and discuss the recent work from Ehrenhofer-Murray (Anedchenko et al., 2019). This group found the same Cse4-Okp1 interaction and has also described Cse4 modifications that influence this interaction.

[Editors' note: further revisions were requested prior to acceptance, as described below.]

Thank you for resubmitting your work entitled "Interaction of Sli15/Ipl1 with the COMA complex is required for accurate chromosome segregation in budding yeast" for further consideration at *eLife*. Your revised article has been favorably evaluated by Anna Akhmanova as the Senior Editor, and a Reviewing Editor.

The manuscript has been improved but there are some remaining issues that need to be addressed before acceptance, as outlined below:

1) The chromosome loss assay described in Figure 6 (and the paragraph discussing it) should be removed from the paper. In such an assay, completely red colonies cannot be interpreted as 100% chromosome loss. If colonies are completely red (which they are in this case), it suggests that the strain does not have the minichromosome and that it was lost at some unknown point during the experiment.

2) Based on the removal of the chromosome loss assay, the manuscript title should be revised.

---

## [Author Response]

Essential revisions (experimental):1) The discovery of a potentially new binding site for the CPC at the inner centromere is an important advance, but there is no demonstration that this interaction site exists in vivo, nor what the phenotypes of impairing this interaction are. While the tethering experiments are compelling, the authors "anchor-away" the entire Ctf19 protein, which leaves open the possibility that the phenotypes are due to multiple Ctf19 functions. An attempt to analyze Ipl1 localization in vivo or show that error correction requires the interaction between Ipl1 and Ctf19 would strengthen the findings. Ideally, the authors would use their conditional system to show that the mutation of the Ctf19 binding site eliminates all Ipl1 from kinetochores when the Sli15 N-terminus is mutated. At a minimum, the authors should analyze bi-orientation in a Sli15 mutant in the Ctf19 binding site to show that the cells are dying due to a bi-orientation defect as expected.

We agree with the reviewer that assessing the effect of the perturbed interaction between Ctf19/Mcm21 and Sli15/Ipl1 on kinetochore localization of Sli15/Ipl1, chromosome biorientation and segregation, is inevitable for understanding the functional implications of the in vitro identified binding interface. Indeed, we invested quite some time and effort, even before the initial submission of the manuscript, to monitor Sli15 localization to the kinetochore upon Ctf19 depletion in a sli15ΔN background using IF microscopy.

In January, the Tanaka lab posted a preprint of a related study reporting very similar conclusions by a complementary approach. The Tanaka lab applied a more sophisticated system with the auxin-inducible degradation of Mcm21 and Bir1 and the centromere reactivation assay. By quantifying the Ipl1 signal only at kinetochores that had not been recaptured by microtubules, they clearly showed the effect of the Sli15 N-terminal deletion and degradation of Mcm21 and Bir1 on the kinetochore recruitment of Ipl1.

For these reasons and due to the time constraints, we refrained to reproduce this result. Instead, we used our artificial tethering approach to investigate whether a Ctf19ΔC-Okp1 fusion protein, that causes synthetic lethality in the sli15ΔN background, displays chromosome segregation defects in the SLI15 background. Using a minichromosome segregation assay we found that cells upon anchoring-away endogenous Ctf19 and ectopically expressing Ctf19ΔCOkp1 exhibited 100% segregation defect whereas expressing Ctf19-Okp1 resulted in 31% red/sectored colonies and rescue with wild-type Ctf19 displayed 22% red/sectored colonies. The deletion of the Sli15 in vitro binding site on Ctf19 thus correlates with a strong segregation defect in the SLI15 background (Figure 6).

2) It is difficult to see some of the proposed interactions in Figure 4B -- quantification of this binding experiment would be useful. An additional issue with this figure, which is crucial to the manuscript as a whole, is that pulled-down material for single inputs alone (Sli15/Ipl1, Ame1/Okp1, Ctf19/Mcm21, etc.) are not shown, making the experiment difficult to interpret. These controls should be included.

We have split up the Figure 4B into 4B, 4C and 4D and included the requested single input lanes in 4B. The experiment in 4C shows the effect of the deletion of RWD-C in Ctf19 on the Sli15 binding of Ctf19/Mcm21 together with the quantification of this experiment in 4D. We did not quantify the experiment in 4B as the phosphorylation/dephosphorylation resulted in shifts and smearing of the Sli15 band which prevented a direct comparison between the three (native, autophosphorylated, dephosphorylated) Sli15/Ipl1 conditions. Ctf19/Mcm21 and Ame1/Okp1 bind Sli15/Ipl1 and the interaction is abrogated upon Sli15/Ipl1 autophosphorylation. Moreover, binding of Ctf19/Mcm21 critically depends on the Ctf19 Cterminal RWD domain. An interpretation beyond these obvious conclusions requires additional experiments with a different set up to assess Sli15/Ipl1 interactions in the context of larger kinetochore assemblies.

3) The authors demonstrate that phosphorylation of Sli15 by Ipl1 negatively regulates CPC binding to Ctf19, but there is no indication that this is physiologically relevant. If possible, the authors should test if the phospho-mutants (or mimics) have an effect on viability or chromosome segregation.

We tested the Sli15 phosphorylation-mimicking mutants indicated in the (original) Figure 4C, 4D and deletion mutants of the P4 (aa 350-490) region for cell growth, benomyl sensitivity and for their ability to segregate a centromeric plasmid. The phenotypes of the different mutants were very subtle and did not support functional conclusions or further interpretation. In addition, we repeated the in vitro binding assay with an extended set of mutants in order to assess the Ctf19 binding site on Sli15. This strategy did not result in a reproducible identification of a distinct Sli15 region mediating the Ctf19 interaction. Due to the subtle phenotypes and the ambiguous results of the in vitro binding assay we have decided to withdraw the Figure 4C and 4D from the manuscript. We think that characterization of the Ctf19 binding site on Sli15 will require further screening with an extended set of mutants which eventually will identify mutants for the functional analysis of their implications in chromosome biorientation and segregation.

4) Related to the above point, the authors expressed Sli15-Ipl1 in insect cells and show basal phosphorylation. Might there be some level of phosphorylation that impacts COMA interaction before ATP treatment? It would be informative to test if the interaction is positively affected after treatment with lambda phosphatase.

As suggested by the reviewer we have included Sli15/Ipl1 that was dephosphorylated on beads by lambda phosphatase prior to incubation with Ame1/Okp1 and Ctf19/Mcm21. We did not monitor the dephosphorylation step by mass spectrometry, but SDS-PAGE and Coomassie staining showed a more focused band of Sli15 upon dephosphorylation and the levels of the bound proteins did not grossly change upon dephosphorylation.

5) Figures 2D, 5D and 5E: Immunoblots to demonstrate protein expression should be included.

We have included the requested immunoblots in the respective panels.

Essential revisions (non-experimental):1) "Epigenetically marked" is inappropriate for budding yeast.

We have changed the phrase accordingly.

“Kinetochore assembly is restricted to centromeres, chromosomal domains that are marked by the presence of the histone H3 variant Cse4^CENP-A^”.

2) Introduction, third paragraph: The end of this paragraph needs to be rewritten. The authors say that Campbell showed Sli15dN could still localize to kinetochores. They then discuss centromere-targeting deficient Sli15/Ipl1. Which is it? If the authors are referring to distinct Sli15/Ipl1 pools (kinetochore versus centromere?), then they should make the reference to distinct pools explicit and substantiate them with the proper references.

Here, we described the findings of Campbell and Desai, in particular, the altered localization pattern of Sli15 upon deletion of the N-terminus. In budding yeast, CPC is recruited through Ndc10 and through the localization of shugoshin to H2A phosphorylated by Bub1 whereas the latter has not been fully experimentally established in budding yeast (to our knowledge). Whether these two recruitment pathways result in distinct CPC pools with distinct functions and localizations in early mitosis remains to be shown. Moreover, H2A bound shugoshin and Ndc10 may be localized in very close proximity at centromeric nucleosomes raising the possibility that the scaffold of Sli15^INCENP^ extends to and positions Ipl1 at the kinetochore structure.

We do not see the contradiction in referring to Sli15ΔN as centromere-targeting deficient and kinetochore-localized at the same time. We agree with the reviewer that the alternating use of inner centromere, centromere or kinetochore localization is confusing. We have rephrased the paragraph.

For the description of our data we have thus used 'centromere/inner kinetochore' localization for Sli15 wild-type and 'inner kinetochore' localization for Sli15ΔN according to the overall localization patterns reported by Campbell and Desai and the observation that Ame1 has a role in Sli15 'kinetochore-localization' by Knockleby and Vogel, 2009.

“A recent study by Campbell and Desai challenged this model by showing that a mutant version of Sli15 lacking the centromere-targeting domain, Sli15∆N2-228 (Sli15∆N), suppresses the deletion phenotypes of Bir1, Nbl1, Bub1 and Sgo1 that mediate recruitment of the CPC to the centromere. […] These findings suggest that centromere-targeting of Sli15/Ipl1 is largely dispensable for error correction and SAC signalling. But a molecular understanding of how the inner-kinetochore localised Sli15∆N/Ipl1 retains its biological function is missing.”

3) Introduction, last paragraph: Topology is the wrong word.

We have changed the phrase accordingly.

“We describe here the use of chemical crosslinking and mass spectrometry (XLMS) together with biochemical reconstitution to characterize the CTF19c^CCAN^ subunit connectivity and the protein interfaces that establish a selective Cse4-NCP binding environment”.

4) "The interaction is not mediated by AT-rich DNA sequences" should say, "The interaction does not require AT-rich DNA sequences."

We have changed the phrase accordingly.

“The lack of interaction with H3-NCPs, which were reconstituted using the same 601 DNA sequence, suggests that Ame1/Okp1 directly and selectively binds Cse4 and that the interaction does not require AT-rich DNA sequences as previously proposed”.

5) Subsection “The Ame1/Okp1 heterodimer selectively binds Cse4 containing nucleosomes”, last paragraph: The authors describe crosslinks between Mif2n and MIND. Did they use the Dsn1-2D mutant? If not, they should comment on the prevalence of these crosslinks relative to expectations. Why do they see them at all?

We used wild-type MTW1c lacking the Dsn1 S240 and S250 phosphomimetic mutations. As outlined in the text, wild-type MTW1 did form a complex with Mif2 and Ame1/Okp1 on SEC which sufficiently served the purpose of the experiment. We did not investigate the effect of Dsn1-2D on the subunit connectivity between MTW1c and the inner kinetochore (Dimitrova et al., 2016).

“In all in vitro reconstitution and XLMS experiments we used wild-type MTW1c purified from *E. coli* and lacking the phosphorylation mimicking mutations of Dsn1 S240 and S250, which were found to stabilize the interaction with Mif2^CENP-C^ and Ame1 but were not required for complex formation on SEC columns (Figure 2C, Figure 1—figure supplement 1)”.

6) Results, subsection “The Ame1/Okp1 heterodimer selectively binds Cse4 containing nucleosomes” last paragraph: "no direct interaction" is not formally correct. See Schmitzberger, 2017 supplement.

We have changed the phrase accordingly.

“A direct interaction between COMA and in vitro translated Chl4 was reported previously and the Ctf19/Mcm21 heterodimer was found to be required for the kinetochore localization of Chl4 and Iml3”.

7) The following is not a correct conclusion: "Chl4, in contrast to its human ortholog, does not recognize Cse4-NCPs." The strongest claim that can be made is that the authors have not found evidence supporting this idea. Nor do the published XLMS data for the human ortholog provide such evidence -- only direct binding and a structure. One could imagine a number of reasons for absence of crosslinks: no lysine pairs accessible to BS3 link Chl4 and Cse4; 601 is an inappropriate DNA substrate to use for these studies; post-translational modifications are required for Chl4-Cse4 engagement. In short, although the data are clear on many points and extremely valuable overall, they cannot prove the negative.

We agree with the reviewer and have changed the phrase accordingly. We also adapted the respective paragraph in the Discussion (see Essential revisions (non-experimental) 19) in order to avoid an overstatement of the experimental results.

“In contrast to the EMSA titration of human CCAN complexes with CENP-A-NCP using 10 nM NCP mixed with up to 20-fold excess of the respective complex, we could not detect Cse4-NCP band shifts with Chl4/Iml3, the orthologs of human CENP-NL, and with Mcm16/Ctf3/Mcm22, the orthologs of human CENP-HIK (no *S. cerevisiae* ortholog of CENP-M has been identified) using 500 nM NCP mixed with a twofold excess of the CTF19c subcomplexes”.

8) "END domain" is redundant.

We have corrected this redundancy in the entire manuscript.

9) "suggests that recruitment of the Ame1/Okp1 heterodimer" This should be rewritten to clarify the point that the suggested recruitment relationship between Cse4-END and Okp1-Ame1 is the speculation here. This modification is important; although the evidence presented is consistent with recruitment (Okp1-Ame1 by Cse4), no conclusive test of such a relationship is presented. An alternative hypothesis – that Cse4-END regulates an essential function of either Okp1-Ame1 or Cse4 – is equally likely given the data.

We agree with the reviewer and have changed the phrase accordingly.

“The observation that deletion of the minimal Ame1/Okp1 interacting Cse4 motif (aa 34-46) correlates with the loss of cell viability, whereas the C-terminal half of the END (aa 48-61) is neither essential for viability nor required for Ame1/Okp1 association suggests that binding of the Ame1/Okp1 heterodimer to Cse4 residues 34-46 is essential for yeast growth”.

10) "Ctf19 is the primary Sli15/Ipl1 interaction site […]" This statement is too strong. Instead, it should read "is required for Sli15/Ipl1 interaction"

We agree with the reviewer and have changed the phrase accordingly.

“In agreement with previous findings recombinant Ctf19ΔC formed a stoichiometric complex with Mcm21, but lost its ability to bind Sli15/Ipl1 indicating that the RWD-C domain of Ctf19 is required for Sli15/Ipl1 interaction with Ctf19/Mcm21 in vitro (Figure 4B)”.

11) Subsection “Tethering Sli15ΔN selectively to COMA rescues the synthetic lethality of a sli15ΔN mutant upon Ctf19 depletion”, last paragraph: This paragraph should be split into at least two and possibly three or four paragraphs.

We split up this section into two paragraphs.

12) Subsection “Tethering Sli15ΔN selectively to COMA rescues the synthetic lethality of a sli15ΔN mutant upon Ctf19 depletion”, last paragraph: That Ctf19dN is not synthetic with Sli15dN is both interesting and important. If impaired cohesin loading does not explain the synthetic interaction with Ctf19d (as apparently it does not), how would the authors interpret the finding from Campbell and Desai that Ctf19d is synthetic with Sli15dN? When this finding was originally presented, it was framed as evidence for the cohesin hypothesis, which here seems discredited. A single sentence in the Discussion section would be appropriate.

We have included the following sentence.

“Moreover, assessing the initially proposed model by Campbell and Desai (Campbell and Desai, 2013) for the synthetic growth defect of Ctf19/Mcm21 deletion in a sli15ΔN background, we observed that deletion of the Ctf19 N-terminus, previously shown to be required for recruiting the cohesin loading complex Scc2/4 (Hinshaw et al., 2017), did not cause a synthetic effect in sli15ΔN mutant cells. This result indicated that the synthetic effect with the Sli15 N-terminal deletion is mediated by a Ctf19 domain distinct from its N-terminus and its role in cohesin loading”.

13) "Endogenous Sli15dN could not rescue in trans […]" and "In contrast, deletion of the SAH domain in Ame1- and Okp1-Sli14dNdSAH fusions was not lethal and could be rescued in trans." These statements are vague. Could not be rescued by what in trans? I suspect this is a reference to later Campbell work (Fink et al., 2017) showing that induced dimerization of Sli15 rescues viability of cells expressing more perturbative Sli15dN alleles (d2-500, for instance). If so, this should be made explicit and the reference included here. Even so, I don't quite see how the current findings address the later Campbell observations in any substantial way.

We agree that these statements are vague and thus, have rephrased the paragraph as indicated below. Indeed, we did not intend to address the Fink et al. 2017 observation that dimerization rescues the otherwise lethal N-terminal truncations of Sli15.

“Hence, since the ectopically expressed fusion proteins were tested in the sli15ΔN background, the result indicates that Ipl1 activity associated with endogenous Sli15ΔN could not rescue synthetic lethality and that tethering Ipl1 activity to COMA subunits is crucial. In contrast, deletion of the SAH domain in Ame1- and Okp1Sli15∆N∆SAH fusions was not lethal and was presumably rescued by the SAH domain of the endogenous Sli15ΔN protein (Figure 5D) suggesting that the SAH domain is not required for the function of the inner kinetochore-localized CPC pool”.

14) "[…] suggesting that the SAH domain is not required for the function of the CPC proximal to the centromere." What do the authors mean by "proximal to the centromere?" My understanding is that CPC has multiple functions, likely corresponding to distinct modes of localization: (1) KT-bound CPC localizes, as shown here, through COMA; 2) MT/spindle-bound CPC localizes through Sli15 and is regulated by CDK; 3) CPC targets chromatin through a poorly-defined localization pathway that probably differs in yeast and vertebrates and that corresponds to H3 phosphorylation throughout the pericentromere. This third pathway appears to be disabled by the Sli15dN allele, but this property has been a source of some confusion, because whether CPC localizes to the centromere seems to depend on MT attachment status and the cell cycle. The third pool of CPC must sense cell-cycle regulation as it appears to be involved in chromosome condensation. It is unclear which CPC population the authors refer to in the statement quoted above. If it's the third, then they need to look at pH3. If they would rather not address this difficult topic, then the sentence should be rewritten to clarify this point.

We have changed the phrase accordingly. Deletion of the yeast haspin-like kinases ALK1 and ALK2 does not result in a synthetic growth defect in a sli15ΔN background and thus, it is unclear whether the H3-T3-P dependent mechanism operates in yeast (Campbell and Desai, 2013).

“[…] suggesting that the SAH domain is not required for the function of the inner kinetochore-localized CPC pool”.

15) Subsection “Tethering Sli15ΔN selectively to COMA rescues the synthetic lethality of a sli15ΔN mutant upon Ctf19 depletion”, last paragraph: The text references Figure 5E but makes no mention of the benomyl experiment. How do the authors interpret this experiment (see also comments in the summary)?

We attribute this discrepancy between our results and the findings made by Campbell and Desai to different experimental conditions. As we have repeatedly seen the benomyl sensitivity of the Sli15ΔN mutant, we think it is important to include this observation in the manuscript. A recent paper by the Kelly lab (Haase et al., 2017) showing that in *Xenopus* egg extracts CPC lacking the CEN domain of INCENP affected the inner kinetochore composition and the correction of erroneous kinetochore-microtubule attachments. They discussed that the centromere-targeting may be important for establishing a sharp phosphorylation gradient as it has been proposed for the spatial separation model of error correction. But even 'noncentromeric' inner kinetochore-localized CPC may establish a flat gradient and intermediate levels of Ndc80 phosphorylation that prevent attachment errors, especially in yeast with kinetochores attached to one microtubule, unless cells are challenged by microtubule depolymerization.

We have modified the paragraph in order to describe the different experimental set-ups.

“Cells ectopically expressing the Sli15∆N mutant protein grew like wild-type, but displayed sensitivity to 15 µg/ml benomyl which contrasted the observation by Campbell and Desai that cells carrying the endogenous sli15ΔN allele were not sensitive to 12.5 µg/ml benomyl (Campbell and Desai, 2013), hence, these deviating observations may be a result of different experimental conditions”.

16) Subsection “The Ame1/Okp1 heterodimer directly links Cse4 nucleosomes to the outer kinetochore”, first paragraph: condense this paragraph. Don't rehearse all the data: just summarize the results in 2-3 sentences and end with the conclusion stated at the end of the aforementioned paragraph.

We are grateful for the suggestion and have shortened the paragraph.

“We investigated the subunit connectivity of the inner kinetochore assembled at budding yeast point centromeres at the domain level using in vitro reconstitution and XLMS. We found that in addition to Mif2 (Xiao et al., 2017), the Ame1/Okp1 heterodimer of the COMA complex is a direct and selective interactor of Cse4-NCPs and characterized the binding interface between Ame1/Okp1 and Cse4. We identified the residues 163-187 of the Okp1 core domain (Figure 3B, C) (Schmitzberger et al., 2017) and the residues 34-46 (Figure 2D, E) of the Cse4 END (aa 28 to 60), which is conserved between interrelated yeast, to establish the interaction. The notion that the essential function of the Cse4 N-terminus and the binding interface for Ame1/Okp1 are mediated by the same 13 amino acid motif suggests that the interaction of Ame1/Okp1 with Cse4 is essential in budding yeast”.

17) "As MTW1c recruitment by Mif2 and Cnn1 are redundant […]" There are at least two issues with this statement. First, there is no definitive evidence that yeast Cnn1 recruits MIND. There is, however, good evidence that Cnn1 interacts with Spc24/25. Second, although the two Ndc80 recruitment modules (Cnn1N and MIND via Mif2N) appear redundant, they likely serve distinct functions at different times during the cell cycle. Indeed, the two recruitment modules appear to be tightly regulated by phosphorylation, and different sets of kinases regulate the relevant components.

We have rephrased the paragraph as indicated below in order to make a clear statement. We refrain from discussing the different pathways of NDC80 and MTW1 recruitment and their regulation in detail as this goes beyond the scope of the discussion of this study.

“As NDC80c recruitment by Cnn1 and Mif2 via MTW1c are redundant pathways and become important only if one of the two is compromised, [...]”.

18) "Ame1/Okp1 is the sole essential link of the centromeric nucleosomes to the outer kinetochore […]" This statement is too strong. First, the authors have not conclusively shown that the phenotypes they observe are due to impaired Okp1-Ame1 recruitment. Second, the double mutant Cnn1d Mif2dN strain is sick and especially so at elevated temperature (Hornung, 2014). Would this not also be an essential link? Third, Mif2N and Ame1N appear to be two parts of the same connection to MIND. The authors must be explicit about necessary and sufficient conditions here, otherwise the above statement is vague and potentially misleading.

Deletion of the N-terminal 15 amino acids of Ame1 is lethal whereas the Mif2 N-terminus or Cnn1 are not essential for cell growth and the double mutant Mif2ΔN cnn1Δ shows a growth defect which is enhanced at higher temperature (Hornung et al., 2014). The role of Mif2 in linking the outer kinetochore and Cnn1 are thus not essential for cell growth.

As the essential function of the Cse4 N-terminus and the binding interface for Ame1/Okp1 are mediated by the same 13 amino acid motif we describe Ame1/Okp1 as the essential link of nucleosomes to the outer kinetochore. We have modified the text as indicated below.

“Although, we did not address whether the Cse4 residues 34-46 are essential for

Ame1/Okp1 kinetochore recruitment, the notion that the essential function of the Cse4 N-terminus and the binding interface for Ame1/Okp1 are mediated by the same 13 amino acid motif suggests […]”.

“[…] Ame1/Okp1 is the essential link of the centromeric nucleosomes to the outer kinetochore emphasizing its role as a cornerstone of kinetochore assembly at the budding yeast point centromere”.

19) "is consistent with our finding of Chl4/Iml3 being positioned distal from Cse4" The authors have not positioned Chl4/Iml3 with respect to Cse4. Instead, they have not found any conclusive evidence that Chl4/Iml3 and Cse4 are near each other. Surely, Chl4 binds DNA. Would this DNA binding be carried out distal to the Cse4 nucleosome? How distal?

We agree with the reviewer and have modified the text accordingly.

“In contrast to Mif2 and Ame1/Okp1, we did not detect complex formation of Chl4/Iml3 with Cse4-NCPs in our EMSA (Figure 1A). Whether this observation can be attributed to the lack of conservation of the RG motif in the corresponding Cse4 sequences in related budding yeasts (Figure 2A), and whether this reflects a different role of Chl4/Iml3 in Cse4 recognition and kinetochore assembly remains to be determined”.

20) "The distinct architectures of vertebrate and budding yeast inner kinetochores […]" The authors have not shown conclusively that the "architectures" differ substantially. Although they have shown that Cse4/CENP-A engagement may differ, it is unclear by how much they differ, to what extent experimental differences contributed to the perceived divergence, and whether the apparent differences actually reflect differential regulation in yeast and vertebrate cells. The above statement should be amended to reflect this uncertainty. If the authors wish to state that there are substantial differences, they should present this as a hypothesis and not a direct observation arising from their data. In general, this section would benefit from a direct comparison with the crosslinking data presented in the Weir paper (human kinetochore crosslinking mass-spectrometry), of which Herzog was a co-author. An overall assessment of the ways these maps agree and differ would be helpful. In any case, the last three sentences of the subsection “Dual recognition of Cse4 at point centromeres by a CCAN architecture distinct from vertebrate regional centromeres” are unduly speculative, so if the authors do not wish to make the Weir comparison explicitly, they should delete most or all of those sentences, as they distract from the genuinely novel findings about CPC tethering.

We agree with the reviewer that this paragraph is speculative and also redundant to some extent. We also refrain to compare the human (Weir et al.) and budding yeast (this study) crosslink networks as conditions and reproducibility of both experiments cannot be assessed to the extent needed for a direct comparison. Thus, we have deleted the entire paragraph.

21) "Ctf19 or Mcm21 become essential for viability once Sli15 loses its ability to be recruited to the inner centromere" The authors struggle to differentiate between CPC localization modes. In this case, the error is grave, as Campbell and Desai showed that the Sli15dN mutant does indeed localize to centromeres, although less efficiently than Sli15WT. Further, it seems the point of this manuscript is that Ctf19 supports centromeric Sli15 localization in the absence of Sli15N. Conceivably, the authors mean to differentiate between KT and inner centromere localization, but my understanding is that the "inner centromere" (what I would call pericentromeric) localization of CPC is poorly worked out and has not been knowingly experimentally separated from KT-mediated localization.

Campbell and Desai explicitly make the following statement in their paper:

“[…] localization of Sli15 and Ipl1 between sister kinetochore clusters (analogous to the inner-centromere localization in other species) was observed following brief microtubule depolymerization in asynchronously growing cells …. In sli15(ΔNT) cells, localization between sister kinetochore clusters was lost for both Sli15(ΔNT) and Ipl1 […]; instead weak localization was observed coincident with kinetochore clusters […] Thus, the Sli15(ΔNT)–Ipl1 complex supports accurate chromosome segregation without enriching between sister kinetochores in vivo”.

The localization between sister kinetochore clusters was lost for Sli15ΔN and they referred to this localization also as inner centromere analogous in other species. Furthermore, the localization of Sli15ΔN was termed 'coincident with kinetochore cluster' indicating the difference in localization pattern by distinct descriptions.

As outlined above, we have thus used 'centromere/inner kinetochore' localization for Sli15 wild-type and 'inner kinetochore' localization for Sli15ΔN.

We have rephrased the sentence accordingly.

“Moreover, Ctf19 and Mcm21 become essential when loss of the Sli15 N-terminal segment (Sli15∆N) prevents its targeting to the centromere”.

22) Subsection “COMA and its role in positioning Sli15/Ipl1 at the inner kinetochore”, second paragraph: "in trans" This is vague. Just removing these two words would clarify substantially and adding "by endogenous Sli15dN through its SAH domain".

In contrast, deletion of the SAH domain in Ame1- and Okp1-Sli15∆N∆SAH fusions was not lethal and was presumably rescued by the SAH domain of the endogenous Sli15ΔN protein (Figure 5D) suggesting that the SAH domain is not required for the function of the inner-kinetochore localized CPC pool.

23) "[…] that facilitates the selective recognition and destabilization of erroneous microtubule attachments upon loss of tension." This statement is not only unsupported by the data presented, it is at odds with Figure 5F. If Ctf19-bound CPC is the pool that does tension sensing, then the Sli15dN allele should not cause cells to be sicker than WT on benomyl. Instead, Sli15dN, when present as the only nuclear copy of Sli15, does not support viability on benomyl. This suggests the authors have found a CPC function at the inner KT that is distinct from correcting MT-KT attachment errors and that Sli15N is responsible for tension sensing. Does this KT-specific function reflect a role for CPC in kinetochore assembly? This should be discussed in the text. Might this reflect the possibly that distinct CPC populations do Dsn1 phosphorylation and DASH phosphorylation? Is there another CPC substrate at the inner-KT that is important and that we don't yet know about? Finally, with regard to this point, how do the authors reconcile this finding with Supplementary Figure 2C from Campbell and Desai? Those authors did not see a viability defect when they looked at sli15dN on benomyl. Could this reflect differences in the experimental setup?

Here, we also would like to refer to our response to comment 15.

Indeed, we believe that the observed benomyl sensitivity upon nuclear depletion of endogenous Sli15 and ectopic expression of Sli15ΔN is attributed to the different experimental conditions in comparison to the work of Campbell and Desai. The finding that Ame1/Okp1 fused to Sli15ΔN rescued the synthetic lethality of CTF19-FRB/sli15ΔN cells does not contradict the observed benomyl sensitivity of Sli15ΔN as only nuclear copy.

Whether this benomyl sensitivity indicates a role for the centromere-targeting of Sli15 in tension sensing, error correction or a yet unknown function is highly speculative and requires thorough further analysis. Based on the observations of Haase et al. (see also comment 15.) the centromere-targeting domain could be indeed important for tension sensing and/or for establishing a sharp phosphorylation gradient.

We have added the following paragraph to the description of the model in the caption of Figure 7.

“In line with the observed benomyl sensitivity of cells expressing Sli15ΔN as the only nuclear copy, a recent study in *Xenopus* egg extracts found that CPC lacking the CEN domain of INCENP affected the correction of erroneous kinetochore-microtubule attachments (Haase et al., 2017). […] Flat Ndc80 phosphorylation levels could be sufficient for the non-selective turnover of erroneous kinetochore, especially at budding yeast kinetochores which are attached to a single microtubule, unless cells are challenged by microtubule poisons”.

24) "required for cohesin loading" A minor error: the Ctf19 fragment is required to recruit Scc2/4 to centromeres in late G1 but not for general cohesin loading.

We have deleted this passage in the figure legend (Figure 5A) as it is redundant to the main text where we described it as suggested by the reviewer.

25) Figure 1A: The concentration of NCP (and consequently the tested protein) used is not listed anywhere in the manuscript. This also applies to Figure 3B. Usually, EMSAs are done with very low concentrations of the labeled reagent (DNA here) and a range of concentrations for the tested binder. The authors have therefore set these up slightly unconventionally. That's fine, as long as they specify the conditions clearly.

We have mentioned the concentrations in the Materials and methods section and also described the experimental conditions in the text.

“[…] using 500 nM NCP incubated with a twofold excess of the complexes”.

26) Figure 1B: Why are Nkp1/2 apparently so large? Why are Cnn1/Wip1 so small, and where is Cnn1? What is the double band at ~25 kDa for both nucleosome preparations?

The labelling of the different protein complexes was mixed up and has been corrected.

27) Figure 2A: What do the symbols beneath the alignment mean? They are not referenced in the figure legend. The protein shown in the alignment (also for B) needs to be specified in the figure so that a fast reader does not have to consult the legend.

We have indicated the proteins in Figure 2A and 2B and have described the symbols in the figure caption.

28) Figure 2C: What are the "tags" referred to in the figure? These appear to be different preparations than those used in Figure 1, but there is only one type of coexpression vector listed in the supplement.

We have specified the tags in Figure 2C and added a description to the figure legend Figure 1B.

“[…[individual proteins, recombinantly purified from *E. coli*, used […]”.

If the authors remove panel F, as requested below, then panel 2C could be rearranged to make it more readable. Specifically, the Cse4 mutant being tested should be written in larger font, and the cartoons showing each test should be removed.

We have removed Figure 2F and adapted Figure 2C according to the reviewer`s suggestion.

29) Figure 2F: Remove this panel. It confuses and does not add meaningful information. Why are Cse4END and Okp1 both red? Why are structural models shown for both peptides, as no published experimentally-determined structure exists for either?

We agree with the reviewer and have removed this panel.

30) Figure 3A: The protein being shown in the figure needs to be specified clearly in the figure itself. Can the colors for the three lines be either all the same or more different? It's difficult to read as is. Also, I think including the ∆ is unnecessary here, although it is important in panel B.

We have corrected Figure 3 according to the reviewer´s suggestions.

31) Figure 3B: The ladder should be removed.

We have corrected Figure 3B according to the reviewer´s suggestion.

32) Figure 4B: Why does the full COMA pull down less efficiently than Ctf19-Mcm21 alone? Why did the authors not test Sli15-Ipl1 phosphorylation with Ctf19-Mcm21 alone?I would also suggest putting the legend beneath the gels to make the results easier to interpret.

Up to now, we do not have any experimental data explaining the less efficient binding of COMA to Sli15/Ipl1 in comparison to the individual Ctf19/Mcm21 and Ame1/Okp1 complexes. As we have added Ctf19/Mcm21 and Ame1/Okp1 as individual complexes in an equimolar ratio we speculate that under the experimental conditions Ctf19/Mcm21 and Ame1/Okp1 do not completely assemble into the COMA complex and/or that they compete for overlapping binding sites on Sli15 in the absence of other binding partners like the Cse4NCP or Bir^Survivin^ and Nbl1^Borealin^ in vitro. Based on our crosslinking data we focused on the characterization of the Sli15-Ctf19 interaction and did not further address the binding to Ame1/Okp1.

33) Figure 4D: What is the lower Ipl1 band/fragment, and why is it apparently missing from the pulled-down material for P4, P6, and P7?

We have analysed both protein bands by in-gel digest and mass spectrometry. Both bands have been identified as Ipl1 suggesting that the lower band is a degradation product of Ipl1.

The lower band might have been lost during the washing procedure of the pull-down. Eventually, we have removed Figure 4D what we explain in detail in the response to the essential experimental revisions.

34) Figure 6: Do the authors really believe Mhf1/2 participate in a nucleosome-like particle with Cnn1-Wip1? This should be discussed.

We have corrected the (new) Figure 7 as recruitment of Cnn1/Wip1 critically depends on the localization of the Mcm16/Ctf3/Mcm22 complex (CENP-HIK in humans) (Pekgöz Altunkaya et al., 2016; Basilico et al., 2014).

35) How does this work affect one's reading of Boekmann et al., which describes phosphorylation sites in Cse4N (thought to be Ipl1 sites) and a synthetic interaction between ipl1-ts and mutations at these phosphorylated residues?

We have cited and discussed the work of Boeckmann et al.

36) The authors should cite and discuss the recent work from Ehrenhofer-Murray (Anedchenko et al., 2019). This group found the same Cse4-Okp1 interaction and has also described Cse4 modifications that influence this interaction.

We have cited and discussed the work of the Ehrenhofer-Murray lab.

[Editors' note: further revisions were requested prior to acceptance, as described below.]

Thank you for resubmitting your work entitled "Interaction of Sli15/Ipl1 with the COMA complex is required for accurate chromosome segregation in budding yeast" for further consideration at eLife. Your revised article has been favorably evaluated by Anna Akhmanova (Senior Editor), a Reviewing Editor, and 2 reviewers.The manuscript has been improved but there are some remaining issues that need to be addressed before acceptance, as outlined below:1) The chromosome loss assay described in Figure 6 (and the paragraph discussing it) should be removed from the paper. In such an assay, completely red colonies cannot be interpreted as 100% chromosome loss. If colonies are completely red (which they are in this case), it suggests that the strain does not have the minichromosome and that it was lost at some unknown point during the experiment.

To address the reviewer´s concern that the minichromosome was lost spontaneously during the experiment, we repeated the experiment by plating cells, grown for 4 hours in rich YPD medium, onto synthetic medium in the absence or presence of rapamycin. Colonies on plates lacking rapamycin showed about 3.5 to 7% red/sectored colonies. This demonstrates that the minichromosome is not lost during the experiment and that the minichromosome loss depicted on plates with rapamycin is a consequence of anchoring-away Ctf19 from the nucleus. Furthermore, the rescue of the segregation defect upon the ectopic expression of Ctf19 or Ctf19-Okp1 proteins in contrast to expressing Ctf19ΔC-Okp1 suggested a role of the Ctf19 C-terminus in chromosome segregation.

Indeed, we could not experimentally address the observation that we obtained 100% red/sectored colonies upon Ctf19 depletion. We attribute this to our anchor-away system in combination with possibly suboptimal conditions of the preculture in YPD medium. A thorough optimization of our protocol would have exceeded the time constraints of the revision period.

Apart from this, we think that the current state of the experiment is sufficient to determine the relevance of the Ctf19 C-terminus for chromosome segregation. As we have added the – rapamycin condition, we can clearly show that the minichromosome loss is a consequence of depleting Ctf19 from the nucleus and not due to a spontaneous loss during the experiment. In total, we have repeated the experiment 9 times showing very similar results with low standard errors. Thus, we are confident that the observed effect upon deletion of the Ctf19 C-terminus is valid. In addition, our findings are consistent with the observation of the Tanaka lab (Garcia-Rodriguez et al., 2019) who used a complementary approach showing that removal of Bir1 or Mcm21 reduced localization of Ipl1 at the centromere and that the effect on Ipl1 localization correlated with the establishment of chromosome biorientation.

Furthermore, we agree with the reviewer´s suggestion to revise the text and have outlined the changes in detail in the response to comment 2.

We have avoided any quantitative numerical statements or comparisons based on the results of the minichromosome loss assay and simply interpret the observed changes as effects upon depletion of Ctf19 or deletion of the Ctf19 C-terminus in the Ctf19-Okp1 fusion protein. Moreover, we have removed the phrase “the Sli15/Ipl1 interaction with Ctf19 is required for chromosome segregation” from the text and replaced it with “the Ctf19 C-terminus […] has a role / is important for […] chromosome segregation […]” or similar phrases.

2) Based on the removal of the chromosome loss assay, the manuscript title should be revised.

We have modified the title.

“The COMA complex interacts with Cse4 and positions Sli15/Ipl1 at the budding yeast inner kinetochore”.

We have revised the Abstract.

“Kinetochores are macromolecular protein complexes at centromeres that ensure accurate chromosome segregation by attaching chromosomes to spindle microtubules and integrating safeguard mechanisms. […] This study shows molecular characteristics of the point-centromere inner kinetochore architecture and suggests a role for the Ctf19 C-terminus in mediating accurate chromosome segregation”.

Results:

“Deletion of the Ctf19 RWD-C domain causes a chromosome segregation defect in the Sli15 wild-type background”.

“In contrast, Ctf19∆C-Okp1, which was localized at the kinetochore (Figure 6C), was unable to rescue the segregation defect (Figure 6B, Figure 6B—source data 1)”.

Discussion, subsection “The Ctf19 C-terminus is required for Sli15/Ipl1 binding in vitro and has a role in accurate chromosome segregation”:

“This is consistent with our finding that the Ctf19 C-terminus has a role in accurate chromosome segregation and indicates that the Sli15-Ctf19 interaction contributes to the localization and stabilization of the CPC at the inner kinetochore (Figure 7).”